# Follow-up of bowel endometriosis resections performed using the double circular stapler technique: A decade's experience

Claudio Peixoto Crispi Jr[1,2]*, Claudio Peixoto Crispi[1†],
Claudia Maria Vale Joaquim[1,3], Paulo Sergio da Silva Reis Jr[1,4],
Nilton de Nadai Filho[1,5], Bruna Rafaela Santos de Oliveira[1],
Camilla Gabriely Souza Guerra[1], Marlon de Freitas Fonseca[1,2]

1 Crispi Institute of Minimally Invasive Surgery, Rio de Janeiro, Brazil, 2 Department of Women's Health, Fernandes Figueira National Institute for Women, Children and Youth Health – Oswaldo Cruz Foundation, Rio de Janeiro, Brazil, 3 Department of Proctology, Hospital Federal de Ipanema, Rio de Janeiro, Brazil, 4 Department of Endometriosis, Hospital Universitário Pedro Ernesto – Universidade do Estado do Rio de Janeiro, Rio de Janeiro, Brazil, 5 Department of Obstetrics and Gynaecology, Hospital Ministro Costa Cavalcanti, Foz do Iguaçu, Brazil

☯ These authors contributed equally to this work.
† Deceased.
* claudin.jr@gmail.com

## Abstract

### Study objective

To report individual early and long-term functional outcomes of 43 women who underwent double circular stapler technique (DCST) for colorectal deep endometriosis (DE).

### Methods

This multidisciplinary observational study was a retrospective case series report exploiting a long-established database of clinical information from a single private institution. The cohort consists of consecutive patients from January/2010 through July/2021 who underwent minimally invasive surgical treatment of DE. Inclusion criteria: all women whose bowel DE was managed by DCST. The assessment of bowel function was based on Obstructed Defecation Syndrome score, Gastrointestinal Symptom Rating Scale and Bowel Function in the Community Tool. Outcomes also included intra and postoperative complications, lower urinary tract symptoms, endometriosis-related menstrual and nonmenstrual pain (numeric rating scale), and conception. The analysis of the results was guided by a semi-qualitative reasoning based on individual changes.

**Data availability statement:** All relevant data are within the paper and its Supporting Information files.

**Funding:** The author(s) received no specific funding for this work.

**Competing interests:** The authors have declared that no competing interests exist.

## Results

The follow-up ranged from 1.4 to 123.8 months (median 38.2). All women presented with DE (mostly rASRM stage 4) and underwent large resections. No procedure was converted to open surgery nor required blood transfusion or ostomies. There was no anastomotic leakage. The risk of rectovaginal bowel fistula was 2.3% (CI 95%: <0.1-7.0) – one case. No patient had long-term urinary retention after surgery. At the most recent follow-up on dysuria, dyschezia, dysmenorrhea, dyspareunia and cyclic low back pain, 88 to 100% of women had favorable responses (improvements ≥3 points in symptomatic women or asymptomatic women who remained pain-free). One patient reported important worsening of her intestinal function, requiring continuous use of laxatives. Considering the 20 women with pregnancy intent, 14 (70%) conceived after surgery.

## Discussion / conclusion

Preliminary results were encouraging in the past. The current assessment including long-term follow-up supports DCST for colorectal DE as a feasible, useful, and safe strategy for avoiding segmental colorectal resection when appropriately indicated and properly performed.

## Introduction

### Deep Infiltrating Endometriosis

Endometriosis is an inflammatory condition of multifactorial etiology. It is highly prevalent, affecting roughly 10% of reproductive age girls and women globally (190 million) [1]. Endometriosis frequently causes one or more pelvic dysfunctions, including pain and infertility, which impair a woman's quality of life. [2,3,4]. The most severe form – Deep Infiltrating Endometriosis or just Deep Endometriosis (DE) – refers to endometriosis which infiltrates one or multiple organs to depths exceeding 5 mm [5,6], including the parametrium in the lateral compartment [7] and even pelvic floor muscles [8].

The most common location of extragenital endometriosis is the bowel [9], and the most frequent location of bowel involvement is the rectosigmoid [10]. Gastrointestinal symptoms are nearly as common as gynecological symptoms in women with endometriosis and, curiously, do not necessarily reflect bowel involvement [11,12].

Surgery is still the treatment of choice to improve health-related quality of life, especially in cases in which medical management has been ineffective for pain relief [13,14] or in selected cases of endometriosis-related infertility [15]. To ensure a more complete procedure (avoiding the need for multiple surgeries), large resections may be necessary when multiple deep infiltrating lesions occur and an experienced multidisciplinary team should perform the surgery due to the risk of urinary [16] and bowel [17] complications.

## Implications of the anatomy in nerve-sparing techniques for colorectal endometriosis surgery

When dealing with intestinal endometriosis, pelvic nerves preservation begins with the choice of a conservative resection technique (shaving or discoid). Thus, whenever possible, less invasive procedures should be applied to facilitate preservation of the vascularization and the innervation of the bowel wall [18]. Conservative techniques for bowel resections (shaving and discoid) provide the advantage of minimizing manipulation of the mesorectal tissue, responsible for the vascularization and innervation. Therefore, segmental resection may ultimately be reserved for advanced lesions responsible for major stenosis or for several cases of multiple nodules infiltrating the rectosigmoid junction or sigmoid colon, when conservative resections are not surgically possible [19].

In relation to the innervation of the pelvis, the superior hypogastric plexus ends up to the two hypogastric nerves, which travel along the lateral walls of the rectum before meeting the pelvic splanchnic nerves (S2-S4 roots) to form the inferior hypogastric plexuses. Thus, constituted by a complex tangle of nerves, the inferior hypogastric plexuses contain both parasympathetic and sympathetic fibers. Currently, nerve-sparing surgeries have been systematized with the thought that recognizing nerve structures is necessary for their preservation at the same time as understanding that surgical manipulation is not free risk and unnecessary dissections must be avoided [20, 21]. Some studies suggest the middle rectal artery is an anatomical landmark to discriminate whether to perform a segmental resection of the rectum, without the need to state a recommended distance from the anal margin [18]. Then, when the surgeon is developing the pararectal space, it is also important to identify the middle rectal artery.

Summarily, considering the propositions above, the first step of the systematization in this series has been to identify the ureters bilaterally at the level of crossing with the iliac vessels, followed by the development of the medial pararectal spaces. Secondly, the development of the pararectal space and identification of the hypogastric nerves should be guaranteed. Finally, with the anatomical landmarks recognized, it's possible to decide on the limit of nerve sparing and the radicality of surgery.

## Segmental resection and shaving

Surgery to treat endometriosis that has infiltrated the bowel wall carries unique risks; complications occur in 10 to 25% of these cases [22]. Severe complications occur in about 6% of bowel resections for DE [17]. Segmental colorectal resection using staplers – popularized in the early 1980s – permits a low anastomosis that is safe and technically straightforward [23]. Performing a colorectal anastomosis after a segmental resection using staplers offers some advantages over a traditional hand-sewn anastomosis: 1) there is significantly less contamination, 2) the anastomosis is technically easier to perform, and 3) bowel segments of different diameters can be easily anastomosed [24].

Rectal shaving seems to be associated with fewer postoperative complications than disc excision and segmental colorectal resection [25]. But there is no consensus as to which is preferable; in the hands of experienced surgeons shaving and bowel resection have comparable recurrence rates [26,27].

## Discoid resection

Between the least invasive (shaving) and the most radical techniques (segmental resection with colorectal anastomosis), there is discoid resection for bowel endometriosis. In the past, an open or laparoscopic anterior rectal wall disc excision with a sutured closure was performed on small lesions, which did not require segmental resection. The advent of circular staplers enabled disc resection for specific intestinal lesions [28]. Discoid resection is probably the best option for small lesions due to its feasibility, safety, and low complication rate [29]. However, an immediately performed second discoid resection may be necessary in some special cases in order to completely resect the lesion.

## Complications in bowel resection

A meta-analysis of patients who underwent rectosigmoid resection for endometriosis found that the functional outcomes were comparable between conservative and segmental resection, and that the complication rate did not differ between

discoid excision and segmental resection [30]. However, observational data comparing long-term surgical, clinical, and functional outcomes between conservative and radical surgery in patients with rectosigmoid endometriosis suggest that conservative surgery (shaving, discoid excision) is preferred over radical surgery (segmental resection) in patients with rectosigmoid endometriosis [31]. Actually, further to the direct impact of the intestinal resection technique, the infiltration of the lateral parametrium should be brought into consideration while evaluating functional outcomes [32]. No significant differences in the rate of fistula, bowel leakage, or bladder dysfunction have been observed with segmental excision versus disc resection for colorectal DE, but the risk of bowel stenosis is significantly lower with disc resection [29]. When compared with a nodule excision only, segmental colorectal resection appears to be associated with several unpleasant functional symptoms such as a significant increase in the frequency of bowel movements or severe, albeit rare, post-operative constipation [33]. Avoiding segmental resection whenever possible, spares these complications [34], and achieves better functional outcomes of pelvic organs due to less dissection and greater nerve preservation [35]. The risk of pelvic dysfunctions can be minimized, but not abolished when nerve-sparing techniques are performed [36].

Low anterior resection syndrome (LARS) refers to any persistent alteration of defecation after an anal sphincter-preserving operation for rectal cancer, especially in low and ultra-low resections very close to the anal verge [37]. Among patients undergoing bowel surgery for low rectal DE, LARS is not more frequent after segmental resection when compared with a more conservative approach such as laparoscopic-transanal disk excision [38]. However, whereas nerve-sparing colorectal resection for DE usually preserves sexual function and urinary continence [39], persistent major bowel dysfunction remains problematic.

Even though initial studies did not observe a relationship between the level of the anastomosis and the rate of fistula formation – perhaps attributable to the practice of protective ileostomy in low and ultra-low anastomosis [40] – recent evidence has consistently shown that the complication rate is significantly higher on the left side of the colon, and especially for anastomoses performed within 10 cm of anal verge [13,41,42]. In short: the lower the anastomosis, the higher the risk of problems.

## The double circular stapler technique

The double circular stapler technique (DCST) has been performed by our group for more than a decade in order to avert segmental excision in selected patients, especially in lesions close to the anal verge [43]. The DCST consists of two transanal circular staplers used sequentially to achieve a complete discoid resection of a rectosigmoid lesion. The DCST enables the excision of DE nodules larger than those that can be removed with the single-load technique of the circular stapler. In sum, the DCST is about avoiding a segmental resection in order to reduce the risk of functional impairments potentially caused by surgery [17].

Circumferential nodules or infiltrations responsible for advanced rectal stenosis can rarely be managed by shaving or nodule excision; in other cases, rectal defect resulting from full-thickness excision is so great that correct suture of the digestive wall is impossible, making colorectal resection inevitable [44]. Some approximate landmarks can be used as selection criteria for the DCST: (1) single colorectal lesion is between 2 and 5 cm in length, (2) the lesion affects no more than 40% of the circumference, and (3) the lesion does not obstruct more than 25% of the intestinal lumen [45,46].

Although DCST for bowel DE was described in the international literature as early as 2014 [45], skepticism about the technique has persisted due to the lack of studies that assess pelvic function at both early and long-term follow-up [42]. The aim of this study is to report individual early and long-term functional outcomes of every one of the 43 women who underwent DCST for colorectal DE at a single referral center since 2010. To our knowledge, this is the largest and the most detailed DCST case series reported to date.

## Materials and methods

### Study design

This multidisciplinary observational study is a retrospective case series report exploiting a long-established database of clinical information collected about each of our endometriosis patients. The cohort consists of consecutive patients

referred by their personal gynecologist to the Crispi Institute for Minimally Invasive Surgery – a private institution located in Rio de Janeiro, RJ, Brazil – from January 2010 through July 2021 for consideration of minimally invasive surgical treatment of DE for infertility and/or pain persisting after medical management. The inclusion criteria were all women whose bowel DE was managed by colorectal segmental resection using the DCST. The exclusion criteria were the concomitant use of any other technique to remove a DE lesion in rectosigmoid.

The primary outcome of this study was the absence or presence of complications (both intra and postoperative) of the DCST. The secondary outcomes were the main pelvic symptoms reported at the most recent follow-up assessment. These included changes – that is improvement or deteriorations – in lower urinary tract symptoms, in endometriosis-related pain symptoms, in bowel habits, as well as the reproductive success in those women who tried to conceive, whether naturally or using assisted reproduction techniques (ART).

The outcomes were individually presented and, due to the limited cohort size and the variation in the interval between the original surgery and the most recent follow-up, the analysis of the results was guided by a semi-qualitative reasoning based on individual changes verified in different endometriosis-impacted outcomes. Both the Strengthening the Reporting of Observational Studies in Epidemiology (STROBE) statement [47] and the updated Preferred Reporting of Case Series in Surgery (PROCESS) guidelines [48] were followed (to the extent possible) to strengthen the quality of reporting.

### Ethics

The 43 cases included in this study constitute a subgroup of a larger research protocol, which evaluates the follow-up of DE surgeries performed in patients admitted to the Crispi Institute for Minimally Invasive Surgery. The research protocol was approved on February 21, 2019 by an institutional review board (Research Ethics Committee of the Oswaldo Cruz Institute Foundation-CAAE 07885019.8.0000.5269 IFF-FIOCRUZ). Patients who might have declined to take part in the study would have received the same care as the patients who gave their consent to take part in the study. Starting in January 2018, written informed consent for inclusion in observational studies was obtained prior to surgery from all patients who underwent surgery. Patients whose surgery occurred prior to January 2018 were contacted and invited to participate in the study, and specific written informed consent for inclusion in the study was obtained. If they have not been seen at the Crispi Institute in several years (and were simply followed by their gynecologist), a follow-up survey instrument was administered to update outcomes.

The authors implemented safeguards to protect the confidentiality of the participants throughout all stages of the research cycle. During data abstraction from the medical records, new numerical codes were generated to identify the study cases. No patient identifiers were incorporated into the research database.

### Database and preplanned data collection

The Crispi Institute of Minimally Invasive Surgery has been using standardized instruments and electronic databases to systematically collect and store clinical information for more than a decade. These databases were developed not only to standardize and systematize the clinical documentation of the medical record, but also to foster analytics for quality assurance and to enable future research.

Most of our patients with endometriosis are referred from their regular (continuity) gynecologist. This repository of information includes not only patient demographic data and existing medical comorbidities, but also a detailed "intake" assessment of the principal endometriosis-related symptoms and a detailed description of prior pelvic surgeries performed, including respective videos when available.

A thorough assessment of the main painful symptoms associated with endometriosis is systematically recorded for each patient from the first preoperative visit under strict medical confidentiality. This institutional database has also included results of laboratory tests and reports of diagnostic imaging exams (ultrasonography, computed tomography

scan, magnetic resonance imaging). All evaluations performed by the different specialists of the Crispi Institute's multidisciplinary team and the descriptions of the surgeries performed have been systematically recorded.

Data from both preoperative and immediate postoperative periods (up to 40 days) were obtained by retrospective chart abstraction, whereas data from the surgeries were prospectively collected into a more structured and thus standardized database, including a review of videos of the surgeries. To have the most current outcome information possible, all patients were contacted by telephone, videoconferencing and email between July 2020 and February 2021 to complete the standardized abstraction forms specially designed for this study - an assessment instrument which focused on pelvic pain, lower urinary tract symptoms, bowel function and reproductive outcomes. The retrospective chart abstraction was performed by two experienced nurses with postgraduate training in clinical research. All data were double-checked to ensure maximum accuracy. The data were abstracted from the updated medical records (accessed for research purposes) in 28/02/2021.

### Pelvic symptoms assessment

The severity of the pain was verbally assessed on a self-reported 11-point (0 to 10) numeric rating scale [49]. In order to consider possible variations in pain intensity, study participants were instructed to report their preoperative symptoms as representative of the prior six months. According to the instrument's scale, pain could be hierarchically categorized as none/mild (0-3), moderate/tolerable (4-6), or severe (7-10).

The value of the minimal clinically important differences to assess variations in pain scores could be considered as low as 10% (10 mm on a 100 mm visual scale) [50]. However, this study adopted a more conservative threshold to define a relevant change in pain after surgery; only those variations greater than 3 points were considered clinically relevant [51]. Thus, for each of the pain symptoms assessed, symptomatic women who reported an improvement in pain severity by 3 or more points or asymptomatic women who remained pain-free after surgery (score = 0) were labeled as having achieved a favorable response [52].

Lower urinary tract symptoms were described according to the International Urogynecological Association (IUGA) / International Continence Society (ICS) joint report on the terminology for female pelvic floor dysfunction [53].

The assessment of bowel habits and of the main bowel symptoms was based on the validated Portuguese versions of three instruments: Obstructed Defecation Syndrome score [54], Gastrointestinal Symptom Rating Scale [55] and Bowel Function in the Community Tool [56]. A questionnaire composed of questions most relevant to our patients selected from these three instruments was constructed and administered.

The term "infertility" was used to indicate failure to establish a clinical pregnancy after 12 months of regular, unprotected sexual intercourse [57]. The cyclical symptoms prior to surgery were assessed based on the period in which the patient was under no hormonal blockade. Post-operatively cyclical symptoms were not assessed in those women who had undergone hysterectomy or were under hormonal blockade; for these women the default response was "not assessed".

### The lateral compartment (parametrial region)

The lateral compartment of the pelvis includes the parametrium (which delineates the lateral edges of the uterus), the paracolpium (which delineates the superior third of the vagina and extends laterally into the side walls of the pelvis), and the paracervix (which refers to the inferior aspect of the parametrium, enclosing the pericervical ring) [7].

The parametrium is a richly innervated cellular connective tissue located between the layers of the broad ligament that is commonly infiltrated by DE [3], which can be considered the "neurological electrical unit" of the pelvic viscera; the excision of posterolateral parametrial endometriosis seems to be associated with a higher risk of postoperative dyspareunia and sexual dysfunction compared with women without involvement of the parametria by endometriosis [32].

Large resections in the parametrium (deep parametrectomy), even if carried out by expert surgeons, demonstrate a non-negligible rate of bladder voiding deficit [58]. Furthermore, unilateral nerve preservation during parametrectomy is not sufficient to prevent persistent urinary retention after cytoreductive endometriosis surgery [59].

In this case series, we carefully reviewed resections performed in the lateral compartments, as autonomic nerves that must be protected as much as possible during surgery permeate them. In our detailed description of each of the 43 cases, we employ the 1997 Revision of the American Society for Reproductive Medicine classification of endometriosis (rASRM) [60].

## Surgery

In this case series, the diagnosis of endometriosis involved four steps: medical history, physical examination, magnetic resonance imaging (MRI), and histopathological examination after surgery. The preoperative assessments were conducted on an outpatient basis and the recommendation for surgery was made at the discretion of the Institute's attending gynecologist (C.P.C.). The surgeries were led by the same experienced gynecologist (C.P.C.), who had more than 20 years of experience in DE surgery. The team of specialists who participated in the surgeries (gynecologist, proctologist, urologist, nurse and anesthesiologist) was the same in all cases, with very rare exceptions.

Considering the lack of strong evidences on preoperative bowel preparation for preventing complications in elective colorectal surgery [61,62], the adopted strategies to prepare the patient were determined by the surgeon in charge (Dr C.P.C.) based not only on scientific publications, but also on the professional experience of his multidisciplinary team. Objectively, no important changes have occurred in the last decade regarding the routine strategy. The preoperative bowel preparation is a low residue diet and enemas for rectal cleansing (i.e., sorbitol + sodium lauryl sulphate solution) are used 2 to 4 hours before surgery. Intravenous broad-spectrum antibiotics with action against gram negative and anaerobic community germs (usually amoxicillin with clavulanate or ampicillin with sulbactam unless the patient was allergic to penicillin) were administered immediately before the anesthesia and maintained for 24 hours only.

A combined regional-general anesthetic protocol (nonrapid sequence of induction and intubation) was used. First, spinal anesthesia was performed with isobaric Bupivacaine and Morphine. General anesthesia was then induced with intravenous Propofol (bolus), Alfentanil (bolus) and Rocuronium Bromide (continuous infusion when robotic surgery), and maintained with inhalation Sevoflurane. Multimodal approaches to both pain and postoperative nausea and vomiting were systematically adopted.

With the patient in the Lloyd-Davies position (whether robot-assisted or not) the endometriosis lesions previously identified by physical examination and MRI were sought during laparoscopic exploration of the abdominal cavity and resected. Intraoperative cystoscopy was performed in all patients to explore for and address complex endometriotic lesions involving the bladder or ureter. Hysteroscopy to address intrauterine conditions and the treatment of other diseases not associated with DE (i.e., cholecystectomy, hernia repair) were also performed intraoperatively in some cases, as reported individually in the results section. Every removed specimen was handled carefully and transferred to the pathology department.

For reasons inherent to surgical practice (which is explained by the size of the structures and instruments involved), there are some obvious situations in which DCST is not possible and segmental resection becomes necessary. They include multifocal rectosigmoid lesions (the resection of several segments would be necessary), very extensive lesions (DCST would not be able to remove the entire lesion) and marked stenosis that do not allow the passage of the transanal stapler. Regarding the dimensional criteria for deciding on a double discoid resection, these should be defined by the surgeon, since the size estimates obtained in preoperative imaging exams do not offer millimetric accuracy. In fact, the intestinal nodule is not measured during surgery, but rather its size is compared to the dimensions of the stapler (objectively, the groove created between the anvil and the stapler, when it is open). With a second circular stapler readily available for use, the surgeon performs the first stapling always with the aim of removing the largest part of the lesion (if possible, the entire nodule). Then, a second stapling is performed, if necessary. The current decision flowchart and eligibility criteria for the different surgical techniques for the treatment of bowel endometriosis are shown in **Fig 1**.

All 43 cases in the cohort had DE involving the rectosigmoid portion of the bowel, for which DCST was indicated. The technique performed in all cases was that described in 2014 by our team [45]. Briefly: A systematic approach to the DE

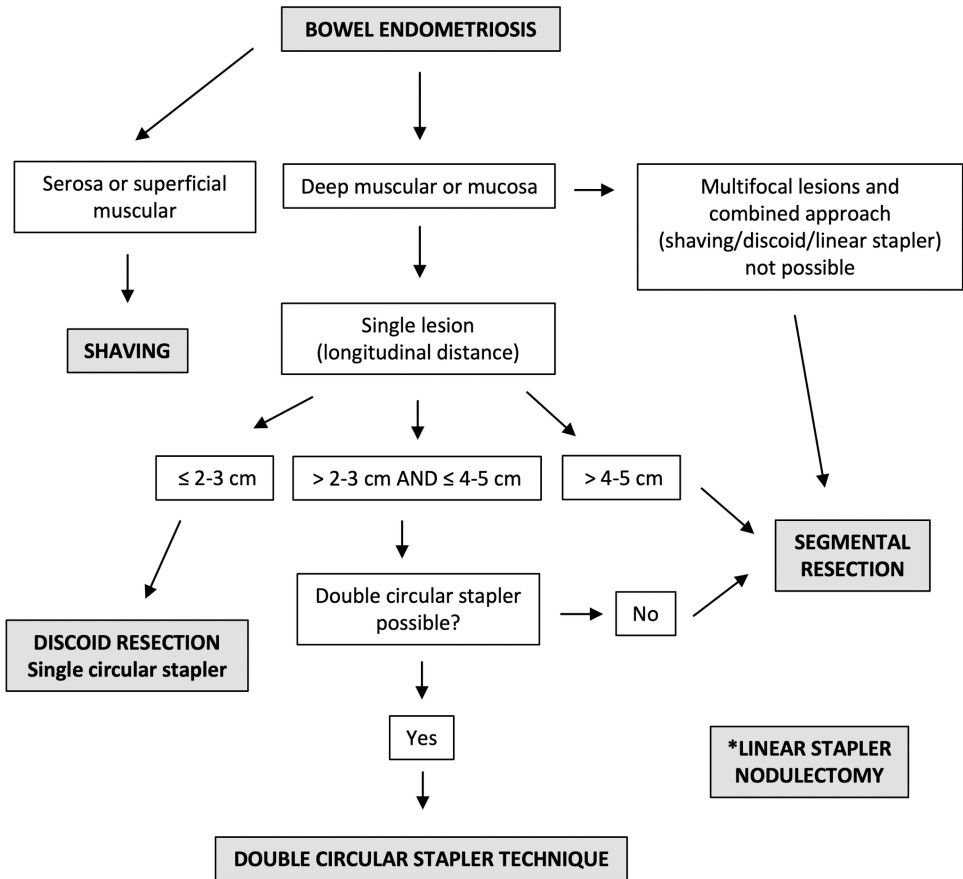

**Fig 1. Current surgical approach to bowel endometriosis resection adopted by the Crispi Institute for Minimally Invasive Surgery – a private institution located in Rio de Janeiro, RJ, Brazil.** Although the linear stapler resection has been shown as a safe alternative to segmental colorectal resection for endometriotic nodules on the anterior rectal wall when less than one-third of the circumference is affected [63], it was not used in this series.

of the posterior compartment and isolation of the nodule is performed. In the DCST, the first suture is passed from the proximal free edge to the middle of the lesion; the passage of the needle is made and involves all layers of the intestine about 0.5 cm proximal to the endometriotic nodule. The second passage of the needle is made deep in the middle of the nodule using 2-0 polyglactin 910 or prolene sutures. A large circular stapler (33 mm in diameter; *Ethicon Circular Stapler™ CDH33A*) is inserted transanally and carefully opened to a length of 3 cm. The area to be excised is laid in the groove created between the anvil and the stapler, caudally orienting the sutures previously threaded into the lesion. The stapler should be tilted upward to prevent the posterior rectal wall from being included in the stapling and to minimize damage to the posterior wall mucosa. The circular stapler is closed and fired (20 seconds after closing), and then removed through the anus. The result is an anterior discoid resection of a wedge of the rectum containing part of the nodule and the suture. The procedure is then repeated, applying the first pass of the needle 0.5 to 1 cm proximal to the stapled free edge and the second pass of the needle at 0.5 to 1 cm from the free distal portion of the endometriotic lesion including all remaining disease and the first stapled line in the second circular stapling using the same first circular stapler size (33 mm) or a smaller one (29 mm-*Ethicon Circular Stapler™ CDH29A*). Usually the circular stapler of smaller diameter is used in the second circular stapling. All these steps are shown in **Fig 2**.

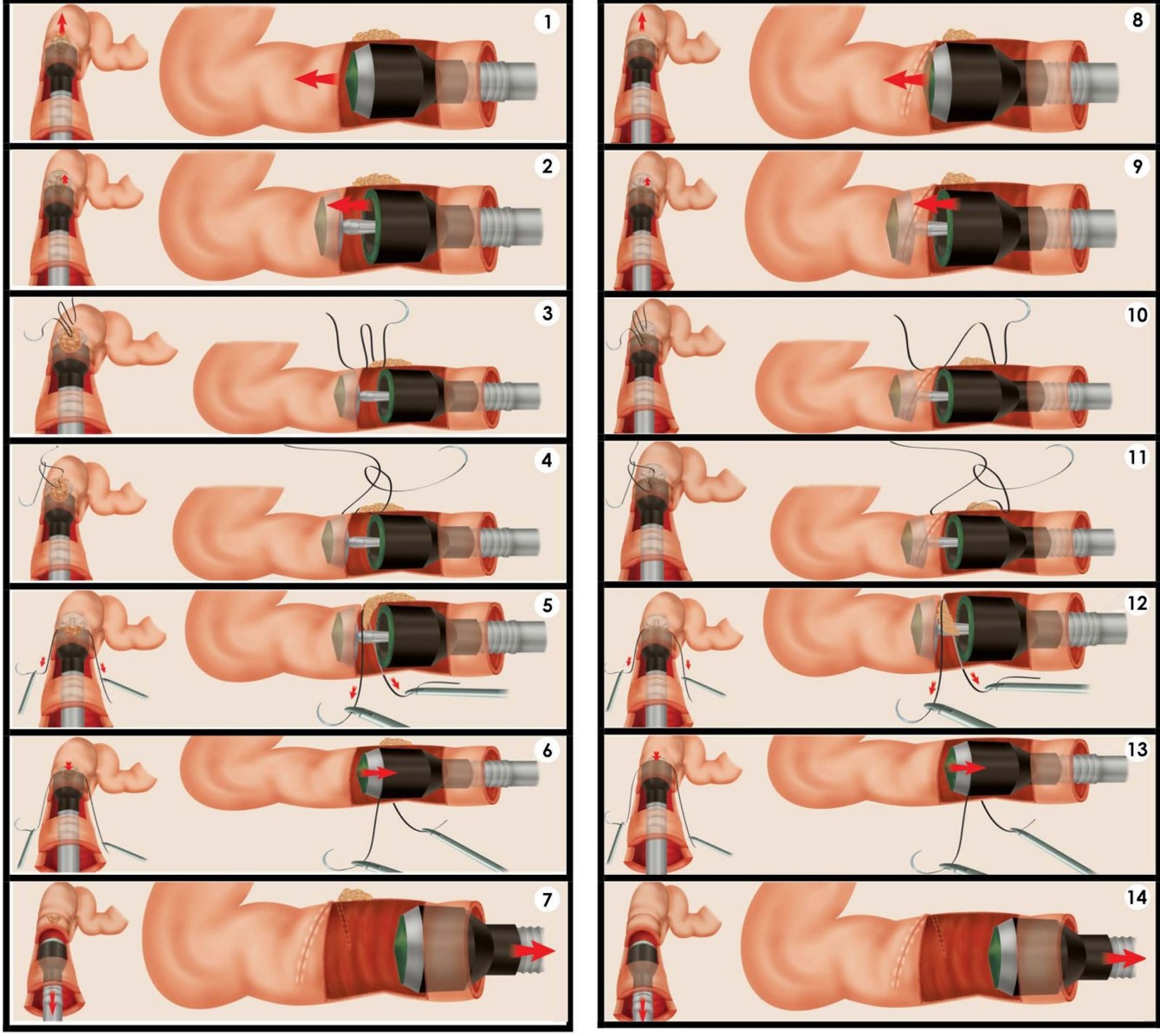

**Fig 2. Main steps of double circular stapler technique for colorectal deep endometriosis.** (1) The circular stapler 1 passes the lesion; (2) Opening the circular stapler; (3) Suture stitch limiting the area to be resected; (4) Suture to anchor the lesion; (5) Sutures forcing the lesion into the stapler; (6) Circular stapler closing; (7) Removing the circular stapler 1; (8) The circular stapler 2 passes the residual lesion; (9) Opening the circular stapler; (10) Suture stitch limiting the area to be resected; (11) Suture to anchor the lesion; (12) Sutures forcing the remaining lesion into the stapler; (13) Circular stapler closing; (14) Removing the circular stapler 2.

We have adopted five surgical strategy guidelines that are applied rigorously (rare exceptions are made): 1) do not insert abdominal drains; 2) do not patch the epiploon; 3) apply biological fibrin glue in the stapling line; 4) perform a stapling line reinforcement using figure-of-eight stiches with polydioxanone 3-0 suture; and 5) perform an anastomotic leak (air insufflation) test.

## Postoperative care protocol

We favor knee-high (18 mmHg) compression stockings (applied immediately before surgery) over pharmacological measures to prevent deep vein thrombosis. Compression stocking are maintained until hospital discharge and patients are encouraged to ambulate as soon as possible. Intermittent pneumatic compression is applied during surgery and is maintained until ambulation.

In all cases, a liquid diet with no residue was reintroduced early, usually the morning after surgery and progressed gradually based on individual acceptance. The timing of definitive bladder (Foley) catheter removal has evolved; the Foley catheter is removed once the residual urine volume is consistently < 100 mL. In addition to clinical evaluation, criteria used for hospital discharge also include non-specific laboratory blood tests to screen for anastomotic leakage, mainly white blood cell count and C-reactive protein concentration.

All patients are followed after discharge by the same multidisciplinary team – which includes a psychologist, a physical therapist (at least up to discharge), a nurse and a nutritionist – typically up to the sixth month, when they usually resume care with their regular gynecologist.

This study included several cases with severe DE conditions that could not be deferred and thus were performed during a period characterized by extreme concern about the risks inherent to the COVID-19 pandemic. Rigorous safety measures were adopted to prevent the healthcare team and patients from becoming infected by the SARS-CoV-2 [64].

## Surgery complications

Complications of surgery were graded according to the Clavien-Dindo classification, where grades I-II are considered minor and grades III-V are considered major [65]. When present, important problems were highlighted and detailed for each patient. The main intra and postoperative complications considered in this study were: conversion to laparotomy, bleeding that required blood transfusion, reoperation, bowel fistula, anastomotic leakage or infections, need of a colostomy or ileostomy, and prolonged dependence on a urinary catheter due to a failure to attain normal control of spontaneous diuresis.

## Statistics

As a descriptive study, this case series report was mathematically limited to calculating the probability of occurrence of some events. For this purpose, the calculation of absolute risk and confidence intervals were performed using IBM SPSS Statistics Version 29.0.0.0 (IBM Corp., Armonk, NY, USA). Although possible in theory, no statistical test comparing before and after outcomes was performed because the duration of follow-up was not standardized.

## Results

### Cohort profile

The study cohort was comprised of healthy Brazilian women who are occasional drinkers, non-smokers, not obese, with a higher education level and middle-class income. Follow-up after surgery ranged from 1.4 to 123.8 months (median 38.2) (**Table 1**). In one case (not included in this study), a segmental colorectal resection was performed because the bowel nodule could not be totally removed by DCST (**Fig 3**).

### Main surgical findings

Suspected endometriotic lesions seen preoperatively on MRI were confirmed by visual inspection during laparoscopy and removed by minimally invasive surgical techniques. The histopathology of tissue specimens collected confirmed

**Table 1. Individual characteristics prior to double circular stapler bowel surgery and at follow-up (prior to/ at follow-up).**

| Case | F-Up | Ethnicity | Mena | Age | Height | BMI | Schooling | Smoking | Alcohol intake | Phys | Main conditions |
|------|------|-----------|------|-----|--------|-----|-----------|---------|----------------|------|-----------------|
| I | 1.4 | Caucasian | 13 | 40/ 40 | 151 | 25.4/ 25.9 | C/ C | never/ never | never/ never | 3/ 3 | no/ no |
| II | 1.5 | Caucasian | 12 | 38/ 38 | 167 | 28.3/ 29.4 | PG/ PG | never/ never | never/ never | 4/ 0 | DM+HBP/ DM+HBP |
| III | 1.6 | Mixed | 12 | 47/ 47 | 162 | 32.0/ 29.7 | C/ C | never/ never | never/ never | 4/ 0 | no/ no |
| IV | 2.5 | Caucasian | 12 | 33/ 33 | 162 | 26.3/ 27.4 | C/ C | never/ never | occasional/ occasional | 3/ 3 | no/ no |
| V | 2.6 | Caucasian | 16 | 34/ 34 | 167 | 19.4/ 19.4 | PG/ PG | never/ never | occasional/ occasional | 2/ 3 | no/ no |
| VI | 4.0 | Mixed | 10 | 40/ 40 | 163 | 25.9/ 24.8 | PG/ PG | never/ never | occasional/ occasional | 3/ 0 | Mig/ no |
| VII | 7.4 | African | 12 | 47/ 47 | 157 | 27.6/ 25.6 | C/ C | never/ never | never/ never | 3/ 1 | no/ no |
| VIII | 13.7 | Caucasian | 13 | 34/ 35 | 172 | 29.8/ 30.4 | C/ C | never/ never | occasional/ occasional | 3/ 4 | no/ no |
| IX | 15.1 | Caucasian | 13 | 36/ 38 | 168 | 31.9/ 31.3 | PG/ PG | never/ never | occasional/ occasional | 0/ 4 | no/ no |
| X | 15.4 | African | 14 | 37/ 38 | 158 | 22.8/ 20.4 | PG/ PG | never/ never | never/ never | 0/ 0 | no/ no |
| XI | 16.1 | Caucasian | 12 | 37/ 39 | 176 | 24.2/ 22.0 | PG/ PG | never/ never | never/ occasional | 4/ 2 | no/ no |
| XII | 16.1 | African | 12 | 31/ 33 | 168 | 23.4/ 23.0 | C/ C | never/ never | occasional/ occasional | 3/ 4 | no/ no |
| XIII | 16.6 | Caucasian | 9 | 44/ 45 | 162 | 27.4/ 30.5 | PG/ PG | never/ never | never/ never | 0/ 3 | no/ no |
| XIV | 18.7 | Mixed | 14 | 35/ 36 | 160 | 18.8/ 25.8 | PG/ PG | never/ never | occasional/ 1 year | 3/ 0 | no/ no |
| XV | 19.7 | African | 12 | 39/ 41 | 180 | 23.2/ 23.5 | C/ C | never/ never | occasional/ 1 year | 0/ 0 | no/ AC |
| XVI | 20.4 | Caucasian | 13 | 36/ 38 | 158 | 22.4/ 23.2 | PG/ PG | never/ never | occasional/ occasional | 3/ 4 | no/ no |
| XVII | 20.9 | Caucasian | 11 | 29/ 31 | 161 | 25.1/ 23.2 | C/ PG | yes/ yes | occasional/ occasional | 2/ 3 | hypo/ hypo |
| XVIII | 21.0 | Caucasian | 11 | 33/ 35 | 159 | 35.6/ 32.44 | HS/ C | never/ never | never/ never | 3/ 0 | DM/ DMi |
| XIX | 21.4 | Caucasian | 12 | 25/ 27 | 170 | 18.7/ 19.7 | HS/ HS | never/ never | occasional/ occasional | 1/ 6 | no/ no |
| XX | 21.8 | Caucasian | 13 | 34/ 36 | 163 | 21.5/ 23.0 | C/ PG | past/ past | occasional/ occasional | 0/ 0 | no/ no |
| XXI | 37.7 | Caucasian | 13 | 38/ 41 | 154 | 27.0/ 26.1 | C/ C | past/ past | occasional/ occasional | 3/ 0 | no/ no |
| XXII | 38.2 | Caucasian | 13 | 35/ 38 | 168 | 26.2226.22 | PG/ PG | never/ never | never/ never | 0/ 0 | no/ no |
| XXIII | 43.8 | caucasian | 13 | 35/ 39 | 168 | 23.0/ 23.4 | HS/ HS | never/ never | occasional/ occasional | 0/ 4 | no/ no |
| XXIV | 45.6 | caucasian | 12 | 31/ 35 | 176 | 23.6/ 26.8 | C/ C | never/ never | never/ never | 0/ 0 | no/ Mig |
| XXV | 48.2 | mixed | 12 | 31/ 35 | 169 | 28.0/ 27.7 | C/ C | never/ never | occasional/ occasional | 0/ 4 | no/ no |
| XXVI | 66.1 | Caucasian | 15 | 30/ 36 | 166 | 20.7/ 21.1 | C/ C | never/ never | occasional/ occasional | 2/ 0 | no/ no |
| XXVII | 67.1 | african | 12 | 37/ 42 | 170 | 27.7/ 23.9 | C/ C | never/ never | occasional/ occasional | 0/ 5 | no/ HBP |
| XXVIII | 72.2 | african | 12 | 30/ 36 | 177 | 19.5/ 20.4 | C/ PG | never/ never | occasional/ occasional | 0/ 0 | no/ no |
| XXIX | 73.2 | Caucasian | 16 | 42/ 48 | 164 | 22.3/ 25.3 | C/ C | never/ never | occasional/ occasional | 5/ 4 | no/ no |
| XXX | 76.3 | Caucasian | 13 | 32/ 39 | 154 | 21.9/ 24.9 | C/ C | never/ never | occasional/ occasional | 5/ 3 | hypo/ hypo |
| XXXI | 79.1 | mixed | 12 | 33/ 40 | 168 | 27.6/ 23.4 | C/ C | never/ never | occasional/ occasional | 4/ 3 | hypo/ hypo |
| XXXII | 82.7 | mixed | 15 | 38/ 45 | 165 | 17.3/ 20.2 | C/ C | never/ never | occasional/ occasional | 0/ 0 | no/ HBP |
| XXXIII | 88.6 | african | 12 | 33/ 40 | 175 | 30.4/ 28.7 | C/ C | never/ never | 1 year/ 1 year | 5/ 0 | HBP/ HBP |
| XXXIV | 94.1 | african | 12 | 31/ 38 | 160 | 27.3/ 32.4 | C/ C | never/ never | occasional/ occasional | 0/ 0 | no/ no |
| XXXV | 96.1 | mixed | 11 | 41/ 49 | 165 | 33.1/ 25.7 | HS/ HS | never/ never | never/ never | 0/ 7 | no/ no |
| XXXVI | 100.4 | mixed | 13 | 27/ 36 | 160 | 26.2/ 27.3 | C/ C | never/ never | never/ never | 0/ 0 | no/ no |
| XXXVII | 104.9 | mixed | 9 | 32/ 41 | 163 | 24.1/ 22.6 | C/ C | never/ never | occasional/ occasional | 3/ 0 | no/ no |
| XXXVIII | 111.9 | caucasian | 13 | 40/ 50 | 163 | 22.2/ 23.7 | C/ C | never/ never | occasional/ occasional | 4/ 2 | no/ no |
| XXXIX | 113.8 | mixed | 12 | 36/ 45 | 175 | 26.1/ 26.5 | HS/ PG | never/ never | occasional/ occasional | 3/ 0 | hypo + P/ hypo + P |
| XL | 115.2 | mixed | 13 | 33/ 42 | 174 | 21.1/ 20.5 | HS/ HS | past/ past | occasional/ occasional | 5/ 0 | HBP/ HBP |
| XLI | 116.3 | caucasian | 11 | 32/ 42 | 161 | 22.0/ 23.2 | PG/ PG | never/ never | never/ never | 0/ 0 | hypo/ hypo+IR |
| XLII | 123.6 | caucasian | 12 | 32/ 42 | 162 | 25.5/ 24.4 | HS/ HS | never/ never | occasional/ occasional | 2/ 5 | no/ DM |
| XLIII | 123.8 | caucasian | 12 | 35/ 45 | 158 | 27.6/ 25.6 | C/ C | never/ never | occasional/ 1 year | 4/ 3 | no/ DM |

F-Up: time of follow-up (months). Ethnicity: self-reported. Mena: menarche (age in years). Age (years). Height (cm). BMI: body mass index (Kg.m$^{-2}$). Schooling (highest completed degree) – HS: high-school; C: college; PG: post-graduate. Alcohol intake (times a week) -1 year: hasn't had an alcoholic drink in a year or more. Phys: physical activity frequency (times a week). Main conditions: regular use of specific medications - Mig: migraine; AC: anticoagulant; Hypo: hypothyroidism; DM: diabetes; DMi: insulin use; HBP: high blood pressure; P: prediabetes; IR: insulin resistance.

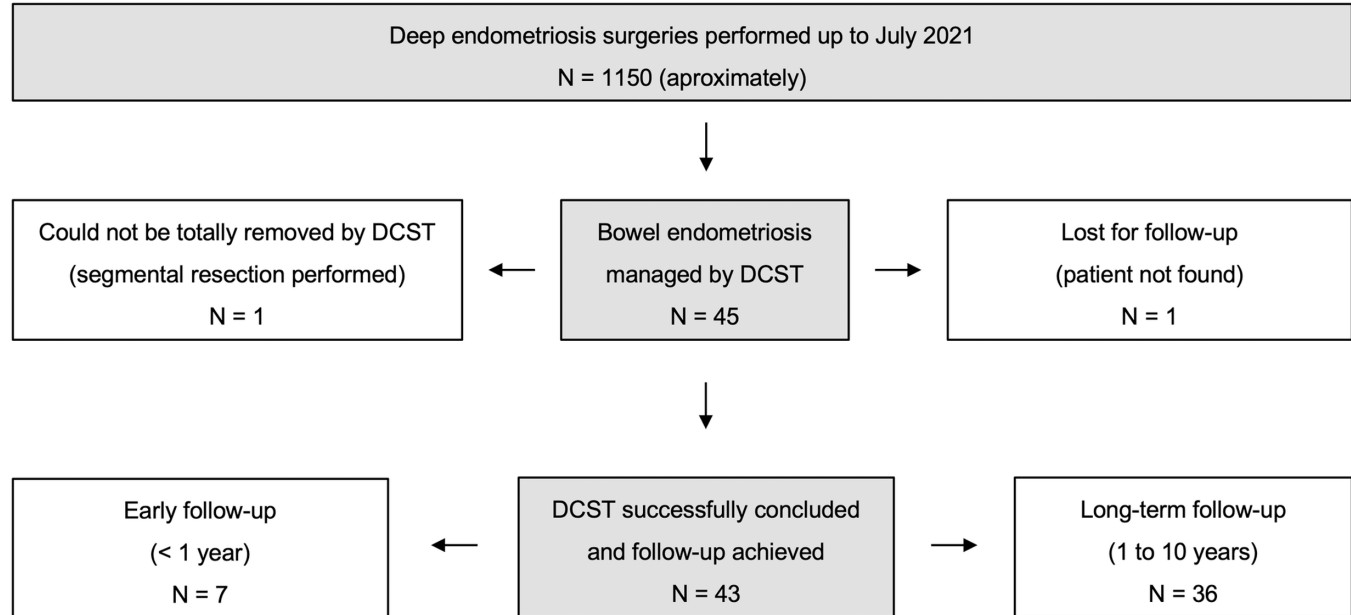

**Fig 3. Study flow diagram-Double circular stapler technique (DCST).** All surgeries were performed by the Crispi Institute for Minimally Invasive Surgery team (Rio de Janeiro, Brazil).

endometriosis in all patients. Overall this cohort of women with rASRM stages 3 and 4 DE (mostly stage 4), presented lesions involving structures such as the ureter, pelvic nerves and the pelvic floor.

Turning to the lateral compartments of the pelvis, 23 patients (53%) had involvement of both sides, which required bilateral parametrial resection for a complete treatment of DE. Of the 43 patients, 2 required nephrectomy due to renal exclusion (assessed preoperatively).

In this series, standardized measurements of the size of the endometriotic bowel lesions were not documented for most of the surgeries and, therefore, this issue could not be accurately explored. Also, accurate and standardized measurements of the distances between the resected endometriotic bowel lesion and the anal verge (determined according to the preoperative MRI report) were not recorded in 15 cases; these distances were thus conservatively documented as "<18 cm" which is the maximum reach of the circular stapler. A detailed written or videorecord of the surgery was not found in three cases (**Table 2**).

The distances from the anal verge were taken from the preoperative MRI report. When not reported, "<18.0 cm", the maximum reach of the circular stapler, was recorded. Lateral compartment: anatomic region with a large presence of pelvic nerves known to be associated with intestinal and urinary functions.

Lap: laparoscopy; Robot: robotic (robot-assisted laparoscopy). Lateral compartment resection: 0: no resection performed; 1: parametrium resected; 2: paracervix resected; 3: paracolpium resected. Other important structures: those subject to a high risk of pelvic nerve injury during surgery.

## Immediate postoperative outcomes and surgical complications

In this series, none of the DSCT procedures needed to be converted to open surgery. None of the cases required a colostomy or an ileostomy. No case required a blood transfusion, and as stated in the Methods section, abdominal drains were not used.

**Table 2. Distances between the endometriotic bowel lesion and the anal verge, and important resections in the lateral compartments of the pelvis performed besides the double circular stapler bowel surgery.**

| Case | Distance (cm) | Surgery | Right/ Left | Other important structures also surgically treated (anterior compartment not included) |
|---|---|---|---|---|
| I | 9.4 | Robot | 1/ 0 | nothing particularly noteworthy to report |
| II | 7.7 | Robot | 3/ 1, 2, 3 | pelvic floor musculature (left) + left hypogastric nerves |
| III | 13.6 | Lap | 1, 2/ 1, 2 | left ureter (extrinsic, without stenosis) |
| IV | 8.6 | Lap | 0/ 2, 3 | ureter bilaterally (extrinsic, without stenosis) + left hypogastric nerves |
| V | 5.0 | Robot | 2, 3/ 2, 3 | ureter bilaterally (extrinsic, without stenosis) + left sacral roots (decompression) |
| VI | 13.3 | Lap | 3/ 1 | nothing particularly noteworthy to report |
| VII | 9.4 | Lap | 1/ 1, 2, 3 | left ureter stenosis (nephrectomy) + right hypogastric nerves + left sacral roots (decompression) |
| VIII | 12.0 | Robot | 1/ 1 | hypogastric nerves bilaterally + left ureter (extrinsic, without stenosis) |
| IX | 11.7 | Lap | 1, 3/ 1, 2, 3 | left ureter (extrinsic, without stenosis) + left obliterated umbilical artery |
| X | 8.9 | Lap | 1, 2, 3/ 1 | pelvic floor (right) + left hypogastric nerves |
| XI | 12.3 | Robot | 1, 2, 3/ 2 | right ureter (end-to-end anastomosis) + right lumbosacral trunk and hypogastric nerves (shaving) |
| XII | 7.4 | Lap | 1/ 0 | right ureter (extrinsic, without stenosis) |
| XIII | 5.2 | Lap | 2, 3/ 0 | right hypogastric/obturator nerves + right ureter (extrinsic, without stenosis) + pelvic floor (right) |
| XIV | 12.7 | Lap | 3/ 3 | left hypogastric nerves + right ureter (extrinsic, without stenosis) + right uterine artery |
| XV | 15.0 | Robot | 2, 3/ 1, 2 | hypogastric nerves bilaterally + right ureter (extrinsic, without stenosis) |
| XVI | 13.3 | Lap | 1, 2/ 0 | right hypogastric nerves |
| XVII | 9.0 | Lap | 1, 2/ 1 | hypogastric nerves bilaterally |
| XVIII | <18.0 | Lap | 2/ 2 | right hypogastric nerves |
| XIX | 11.5 | Robot | 0/ 0 | nothing particularly noteworthy to report |
| XX | 10.2 | Lap | 2/ 2 | nothing particularly noteworthy to report |
| XXI | <18.0 | Lap | 0/ 0 | nothing particularly noteworthy to report |
| XXII | 12.8 | Robot | 1, 2, 3/ 0 | right hypogastric nerves + right ureter stenosis (nephrectomy) |
| XXIII | <18.0 | Lap | 1/ 2, 3 | left ureter (extrinsic, without stenosis) |
| XXIV | <18.0 | Lap | 1/ 1, 2 | left ureter (extrinsic, without stenosis) |
| XXV | <18.0 | Lap | 0/ 2, 3 | pelvic floor musculature (left) |
| XXVI | 7.0 | Lap | 1, 2, 3/ 1, 2, 3 | hypogastric nerves bilaterally + ureter bilaterally (extrinsic, without stenosis) |
| XXVII | 14.2 | Lap | 2, 3/ 1, 2, 3 | pelvic floor musculature (left) + hypogastric nerves bilaterally |
| XXVIII | <18.0 | Lap | 0/ 1, 2 | right hypogastric nerves |
| XXIX | 10.3 | Lap | 2/ 0 | right hypogastric nerves |
| XXX | <18.0 | Lap | 0/ 2 | nothing particularly noteworthy to report |
| XXXI | <18.0 | Lap | 0/ 0 | nothing particularly noteworthy to report |
| XXXII | <18.0 | Lap | 1, 2, 3/ 1, 2, 3 | pelvic floor musculature + hypogastric nerves bilaterally |
| XXXIII | 13.0 | Lap | data not found | data not found |
| XXXIV | <18.0 | Lap | 2, 3/ 0 | right hypogastric nerves |
| XXXV | 12.0 | Lap | 1, 2/ 0 | right hypogastric nerves |
| XXXVI | 8.4 | Lap | 1/ 1, 2, 3 | pelvic floor musculature (left) |
| XXXVII | <18.0 | Lap | 2/ 1, 2, 3 | pelvic floor musculature (left) + left hypogastric nerves |
| XXXVIII | <18.0 | Lap | 2, 3/ 2, 3 | nothing particularly noteworthy to report |
| XXXIX | <18.0 | Lap | 1, 2/ 0 | nothing particularly noteworthy to report |
| XL | 8.0 | Lap | 1/ 1, 2, 3 | left hypogastric nerves + left ureter (with stenosis) |
| XLI | 7.0 | Lap | 0/ 2, 3 | nothing particularly noteworthy to report |
| XLII | <18.0 | Lap | data not found | data not found |
| XLIII | <18.0 | Lap | data not found | data not found |

With regard to postoperative complications, there was no case of anastomotic leakage nor any grade III or IV Clavien-Dindo complication. One patient (case XXXVII) developed a rectovaginal fistula that was clinically diagnosed in the office 12 days after surgery (classified as a grade II Clavien-Dindo complication); this patient was promptly hospitalized and was discharged 15 days later after successful clinical management without the need for reoperation. The absolute risk of such a complication in this series was calculated to be 2.3% (CI 95%: <0.1-7.0).

Five women used a Foley catheter for 30 days or more after surgery due to major endometriosis resections in the urinary tract, which were independent from the bowel endometriosis. No patient in this series experienced long-term urinary retention or needed intermittent bladder catheterization. All women were able to urinate spontaneously after 90 days (Table 3).

One patient whose surgery was performed in 2011 could not be located for contact; no specific reason was identified. Based on the information in her medical record, we were able to establish that she had been discharged from the hospital without surgical complications and without requiring bladder catheterization, that she was evaluated twice postoperatively during outpatient visits at least 30 days apart, and that she had been referred back to the referring gynecologist. After this short-term follow-up she was considered "lost to (long-term) follow-up", though she presumably continued her care with her continuity gynecologist. No additional information is available.

### Urinary findings

Lower urinary tract symptoms were reported by about half of the women both prior to surgery and at the time of the last follow-up assessment. Subjectively, 12 patients (28%) reported improvement of the quality-of-life related to voiding symptoms after surgery, 11 (26%) reported worsening, and 17 (40%) had no noticeable changes. Both menstrual and nonmenstrual dysuria (pain, burning or other discomfort during micturition) was not frequent in this cohort, especially after surgery. It was not possible to discern any clear pattern or trend in relation to changes in urinary symptoms because of the variation in the time elapsed between surgery and the last follow-up (Table 3).

### Pain

Before surgery, pain was the main complaint in 35 women (81% of the patients), but the prevalences of different painful symptoms were quite heterogeneous. Of the 43 women, two-thirds (N = 28) reported some degree of menstrual dyschezia before surgery (scores > 0), with half of these (N = 14) reporting severe menstrual dyschezia (scores > 7). Some degree of nonmenstrual dyschezia (scores > 0) had been reported by 12 women (28%), but only 2 women (5%) had reported severe nonmenstrual dyschezia (scores > 7). All but three women (93%) reported dysmenorrhea before surgery (scores >0), and more than two-thirds of them (72.5%) reported severe dysmenorrhea (scores > 7). Some degree of deep dyspareunia (scores > 0) had been reported by 22 patients (51%); 11 (26%) reported severe deep dyspareunia (scores > 7). Seven women (16%) reported no cyclic lower back pain (scores = 0); 23 women (54%) reported severe cyclic lower back pain (scores > 7). Twenty-six patients (60%) reported some acyclic pelvic pain (scores > 0); 10 women (23%) reported severe acyclic pelvic pain (scores > 7).

Although cyclic symptoms were not evaluated in the women who were undergoing hormonal blockade, major improvements in pain scores after surgery were observed in most cases, regardless of the duration of follow-up (Table 4). At the most recent follow-up on dysuria, dyschezia, dysmenorrhea, dyspareunia and cyclic low back pain, 88 to 100% of women showed favorable responses (improvements ≥3 points in symptomatic women or asymptomatic women who remained pain-free).

### Bowel habits

Regardless of the time elapsed since the surgery, 4 patients (9%) reported some (even minimal) worsening in the quality-of-life related to bowel function at the follow-up assessment, with only one patient (case XXI) reporting severe

**Table 3. Lower urinary tract symptoms prior to double circular stapler bowel surgery and at follow-up (prior to/ at follow-up).**

| Case | F-up | T-Cath | T-int | QoLife | M-Dysuria | NM-Dysuria | Urgency | Intermittency | Feeling | Straining |
|------|------|--------|-------|--------|-----------|------------|---------|---------------|---------|-----------|
| I | 1.4 | 1 | 2 | medium/ good | 5/ n.a. | 4/ 0 | no/ no | no/ no | no/ no | no/ no |
| II | 1.4 | 6 | 6 | medium/ poor | 0/ n.a. | 0/ 0 | no/ no | no/ no | no/ no | no/ yes |
| III | 1.6 | 2 | 2 | excellent/ excellent | 0/ n.a. | 0/ 0 | no/ no | no/ no | no/ no | no/ no |
| IV | 2,5 | 10 | 3 | excellent/ good | 0/ 0 | 0/ 0 | no/ no | no/ no | yes/ no | yes/ no |
| V | 2,6 | 5 | 5 | excellent/ excellent | 0/ 0 | 0/ 0 | no/ no | no/ no | no/ no | no/ no |
| VI | 4,0 | 13 | 3 | excellent/ excellent | 0/ 0 | 0/ 0 | no/ no | no/ no | no/ no | no/ no |
| VII | 7,40 | 20 | 2 | excellent/ poor | 0/ n.a. | 0/ 0 | no/ no | no/ yes | yes/ yes | no/ yes |
| VIII | 13,7 | 1 | 5 | excellent/ good | 0/ 0 | 0/ 0 | no/ no | no/ no | no/ no | no/ no |
| IX | 15,1 | 1 | 4 | good/ good | 0/ 0 | 0/ 0 | no/ yes | no/ yes | yes/ yes | no/ yes |
| X | 15,4 | 7 | 5 | good/ good | 0/ 0 | 0/ 0 | no/ no | no/ no | no/ no | no/ no |
| XI | 16,1 | 1 | 5 | medium/ good | 5/ n.a. | 0/ 0 | yes/ no | no/ no | yes/ no | yes/ yes |
| XII | 16,1 | 1 | 3 | excellent/ excellent | 0/ n.a. | 0/ 0 | no/ no | no/ no | no/ yes | no/ no |
| XIII | 16,6 | 4 | 4 | medium/ good | 0/ n.a. | 0/ 0 | no/ no | no/ no | yes/ yes | yes/ no |
| XIV | 18,7 | 1 | 3 | good/ excellent | 0/ 0 | 0/ 0 | no/ no | yes/ no | yes/ no | yes/ no |
| XV | 19,7 | 10 | 3 | excellent/ good | 0/ 0 | 0/ 0 | no/ no | no/ no | no/ yes | no/ yes |
| XVI | 20,4 | 10 | 5 | good/ good | 0/ 0 | 0/ 0 | no/ no | no/ yes | no/ no | no/ no |
| XVII | 20,9 | 2 | 4 | good/ good | 0/ 0 | 0/ 0 | no/ no | no/ no | no/ no | no/ no |
| XVIII | 21,0 | 1 | 3 | poor/ good | 0/ 0 | 0/ 0 | yes/ no | yes/ no | yes/ no | yes/ no |
| XIX | 21,4 | 2 | 5 | terrible/ excellent | 0/ 0 | 0/ 0 | yes/ no | yes/ no | yes/ no | yes/ no |
| XX | 21,8 | 14 | 4 | excellent/ good | 0/ 0 | 0/ 0 | no/ no | no/ yes | no/ yes | no/ no |
| XXI | 37,7 | 9 | 4 | good/ good | 8/ 0 | 8/ 0 | no/ no | no/ no | no/ no | no/ no |
| XXII | 38,2 | **58** | 5 | good/ medium | 0/ 0 | 0/ 0 | no/ yes | no/ no | no/ yes | no/ no |
| XXIII | 43,8 | 15 | 3 | poor/ good | 0/ 0 | 0/ 0 | no/ yes | no/ no | no/ no | no/ no |
| XXIV | 45,6 | 2 | 5 | medium/ medium | 0/ 0 | 0/ 0 | yes/ yes | no/ no | yes/ no | yes/ yes |
| XXV | 48,2 | 2 | 3 | excellent/ excellent | 0/ 0 | 0/ 0 | no/ no | no/ no | no/ no | no/ no |
| XXVI | 66,1 | 30 | 5 | excellent/ terrible | 0/ 0 | 0/ 0 | no/ no | no/ yes | no/ no | no/ yes |
| XXVII | 67,1 | 1 | 5 | good/ medium | 0/ n.a. | 0/ 0 | yes/ no | no/ no | no/ yes | no/ yes |
| XXVIII | 72,2 | 15 | 4 | good/ good | 0/ n.a. | 0/ 0 | no/ no | no/ no | yes/ no | yes/ no |
| XXIX | 73,2 | 2 | 6 | excellent/ excellent | 0/ n.a. | 0/ 0 | no/ no | no/ no | no/ no | no/ no |
| XXX | 76,3 | 3 | 5 | excellent/ excellent | 0/ n.a. | 0/ 0 | no/ no | no/ no | no/ no | no/ no |
| XXXI | 79,1 | 1 | 2 | good/ good | 0/ 0 | 0/ 0 | no/ yes | no/ yes | no/ yes | no/ yes |
| XXXII | 82,7 | 21 | 6 | excellent/ excellent | 0/ n.a. | 0/ 0 | no/ no | no/ no | yes/ yes | no/ no |
| XXXIII | 88,6 | 1 | 12 | good/ poor | 0/ n.a. | 0/ 0 | no/ no | no/ no | no/ yes | no/ no |
| XXXIV | 94,1 | 1 | 5 | medium/ medium | 0/ 0 | 0/ 0 | yes/ no | no/ no | no/ yes | no/ no |
| XXXV | 96,1 | 20 | 5 | terrible/ medium | 8/ n.a. | 5/ 0 | no/ no | no/ yes | no/ yes | no/ yes |
| XXXVI | 100,4 | 30 | 10 | excellent/ good | 0/ 0 | 0/ 0 | yes/ no | no/ yes | no/ yes | no/ yes |
| XXXVII | 104,9 | **90** | 5 | terrible/ good | 10/ 3 | 10/ 3 | yes/ yes | yes/ yes | yes/ yes | yes/ yes |
| XXXVIII | 111,9 | 1 | 2 | good/ excellent | 0/ n.a. | 0/ 0 | no/ no | no/ no | no/ no | no/ no |
| XXXIX | 113,8 | 1 | 10 | terrible/ good | 0/ n.a. | 0/ 0 | yes/ no | no/ yes | no/ yes | no/ yes |
| XL | 115,2 | 30 | 6 | terrible/ excellent | 10/ n.a. | 0/ 0 | no/ no | no/ no | yes/ no | no/ no |
| XLI | 116,3 | 1 | 2 | good/ poor | 0/ 0 | 0/ 0 | no/ no | no/ no | no/ yes | no/ yes |
| XLII | 123,6 | 2 | 4 | good/ good | 0/ n.a. | 0/ 0 | no/ yes | no/ no | no/ no | no/ no |
| XLIII | 123.8 | 5 | 5 | excellent/ medium | 0/ n.a. | 0/ 0 | no/ yes | no/ no | no/ yes | no/ no |

F-Up: time of follow-up (months). T-Cath: Time (duration) of indwelling Foley catheter (in days). T-int: length of stay in the hospital (days). QoLife: self-reported quality of life related to voiding symptoms (terrible, poor, medium, good or excellent). Dysuria: Complaint of pain during micturition (discomfort may be intrinsic to the lower urinary tract or external); M: menstrual (cyclic); NM: nonmenstrual (acyclic); n.a.: not assessed (under hormonal blockade).

*(Continued)*

**Table 3.** (Continued)

Urgency: Complaint of a sudden, compelling desire to pass urine, which is difficult to defer. Intermittency: Complaint of urine flow that stops and starts on one or more occasions during voiding. Feeling (Feeling of incomplete bladder emptying): Complaint that the bladder does not feel empty after micturition. Straining to void: Complaint of the need to make an intensive effort (by abdominal straining, Valsalva or suprapubic pressure) to either initiate, maintain, or improve the urinary stream. The lower urinary tract symptoms were documented according to the International Urogynecological Association (IUGA)/ International Continence Society (ICS) joint report on the terminology for female pelvic floor dysfunction [53]. Case XXII: the patient underwent nephrectomy and used a bladder catheter for 58 days due to resection of a bladder lesion near the right ostium. Case XXXVII: the patient presented rectovaginal fistula on the 12th day and was hospitalized; she was discharged without reoperation after 15 days of treatment with antibiotics, but the indwelling Foley catheter was maintained for 90 days.

constipation despite continuous use of laxatives after surgery. The risk of a clinically significant worsening in intestinal function after surgery was calculated as 2.3% (CI 95%: <0.1-7.0).

Cyclic abdominal bloating was the most frequent symptom prior to surgery in this series; it was reported by 39 patients (91% of the cohort). No conclusion about changes in cyclic abdominal bloating after surgery has been possible because 20 women were under hormonal blockade at follow-up assessment and, therefore, their cyclical symptoms could not be accurately assessed.

The estimated duration of evacuation increased considerably in two patients (cases IV and VII), but the time elapsed since surgery was still relatively short in one of them (2.5 months), so progression over time to a more normal duration was certainly still possible. Of the 30 women who had not reported a feeling of incomplete evacuation prior to surgery, one-third of them (10 women) reported it at their last follow-up assessment, which characterizes a slight but undesirable change (**Table 5**).

## Reproductive outcome

Of the 20 women who tried to conceive after surgery, 14 (70%) were able to get pregnant, but three of these did not progress to term due to ectopic pregnancy or spontaneous abortion. Eleven of the 15 women who had been diagnosed preoperatively with infertility tried to conceive after surgery (four did not try). Of these 11, five became pregnant and gave birth, three naturally without assisted reproductive technology-ART). Six, including one who used ART, did not become pregnant till the time of follow-up (**Table 6**).

## Discussion

### Main findings

This multidisciplinary observational study presents the individual outcomes of 43 consecutive cases of women who underwent DCST for resection of bowel deep endometriosis at a private medical center specialized in minimally invasive surgery. Although the cases were accumulated over a decade and thus follow-up varied substantially (range: 1.4 months to 10 years and 4 months) the collective long-term outcomes establish DCST for resection of rectosigmoid deep endometriosis as a safe, feasible and advantageous technique when appropriately indicated and properly performed. Moreover, the consequences of the DCST on the global bowel function proved to be quite satisfactory, with the calculated risk of a clinically significant self-reported worsening of intestinal function as low as 2.3% (CI 95%: <0.1-7.0).

### Surgical complications

The overall complication rate of bowel resection for DE reported in the literature is approximately 20% with major complications – anastomotic leakage, fistula and severe bowel obstruction – occurring in about 6.4% of cases [17]. Among our 43 cases, there was no instance of an anastomotic leakage or intestinal obstruction, but only one case (2.3%) had a

**Table 4. Pelvic pain and use of hormones prior to double circular stapler bowel surgery and at follow-up (prior to/ at follow-up).**

| Case | F-Up | Main complaint | M-Dysch | NM-Dysch | Dysm | DDysp | CLBPain | PPain | Hormones at follow-up |
|------|------|----------------|---------|----------|------|-------|---------|-------|----------------------|
| I | 1.4 | Pain | 7/ n.a. | 6/ 0 | 5/ n.a. | 6/ n.s. | 8/ n.a. | 5/ 0 | OP/ ESI |
| II | 1.5 | Pain + Infertility | 0/ n.a. | 0/ 0 | 7/ n.a. | 3/ 0 | 7/ n.a. | 0/ 0 | contCOC/ contCOC |
| III | 1.6 | Pain | 0/ n.a. | 0/ 0 | 7/ n.a. | 0/ n.s. | 9/ n.a. | 5/ 0 | no/ no |
| IV | 2.5 | Pain | 0/ 0 | 5/ 4 | 8/ 5 | 0/ 0 | 5/ 0 | 10/ 0 | LRID/ cycCOC |
| V | 2.6 | Pain | 6/ 0 | 8/ 0 | 9/ 6 | 0/ 0 | 0/ 0 | 8/ 0 | no/ no |
| VI | 4.0 | Imag | 0/ n.a. | 0/ 0 | 6/ n.a. | 8/ 8 | 6/ n.a. | 6/ 6 | no/ no |
| VII | 7.4 | Pain | 0/ n.a. | 0/ 7 | 7/ n.a. | 10/ 3 | 8/ n.a. | 5/ 0 | OP/ no |
| VIII | 13.7 | Pain + Infertility | 0/ 1 | 0/ 0 | 8/ 4 | 0/ 1 | 0/ 0 | 0/ 2 | no/ contCOC |
| IX | 15.1 | Pain | 9/ 0 | 0/ 0 | 9/ 0 | 8/ 0 | 8/ 0 | 9/ 0 | cycCOC/ no |
| X | 15.4 | Infertility | 8/ 0 | 0/ 9 | 9/ 0 | 2/ 0 | 7/ 0 | 7/ 9 | no/ no |
| XI | 16.1 | Pain | 8/ n.a. | 0/ 0 | 8/ n.a. | 0/ 0 | 9/ n.a. | 0/ 0 | OP/ contCOC |
| XII | 16.1 | Pain | 0/ n.a. | 0/ 0 | 8/ n.a. | 0/ 3 | 0/ 0 | 0/ 0 | OP/ contCOC |
| XIII | 16.6 | Pain | 10/ n.a. | 0/ 0 | 10/ n.a. | 0/ 0 | 10/ n.a. | 9/ 0 | no/ no |
| XIV | 18.7 | Pain | 0/ 0 | 0/ 0 | 7/ 3 | 10/ 3 | 8/ 3 | 5/ 0 | OP/ no |
| XV | 19.7 | Infertility | 6/ 1 | 4/ 0 | 10/ 3 | 7/ 0 | 7/ 4 | 7/ 0 | contCOC/ no |
| XVI | 20.4 | Imaging | 5/ n.a. | 0/ 0 | 5/ n.a. | 0/ 0 | 5/ n.a. | 0/ 0 | cycCOC/ contCOC |
| XVII | 20.9 | Pain | 8/ n.a. | 0/ 0 | 10/ n.a. | 8/ 0 | 9/ n.a. | 7/ 0 | contCOC/ contCOC |
| XVIII | 21.0 | Pain | 5/ 0 | 7/ 0 | 8/ 0 | 0/ 0 | 8/ 0 | 0/ 0 | contCOC/ no |
| XIX | 21.4 | Pain | 7/ 0 | 6/ 0 | 10/ 3 | 9/ 2 | 8/ 5 | 8/ 2 | contCOC/ contCOC |
| XX | 21.8 | Pain + Infertility | 0/ 0 | 0/ 0 | 10/ 7 | 0/ 0 | 7/ 7 | 0/ 0 | no/ no |
| XXI | 37.7 | Pain | 0/ n.a. | 0/ 0 | 10/ n.a. | 7/ 7 | 10/ n.a. | 10/ 0 | no/ no |
| XXII | 38.2 | Imag | 3/ 3 | 2/ 2 | 0/ 0 | 4/ 4 | 1/ 6 | 0/ 0 | cycCOC/ no |
| XXIII | 43.8 | Pain + Infertility | 10/ 2 | 4/ 1 | 10/ 1 | 10/ 4 | 1/ 1 | 2/ 0 | no/ contCOC |
| XXIV | 45.6 | Pain | 0/ 0 | 0/ 0 | n.a./ n.a. | 10/ 7 | 10/ 6 | 10/ 6 | LRID/ OP |
| XXV | 48.2 | Pain + Infertility | 7/ 0 | 0/ 0 | 10/ 1 | 0/ 0 | 2/ 0 | 0/ 0 | contCOC/ no |
| XXVI | 66.1 | Imag | 3/ 0 | 0/ 0 | 1/ 0 | 0/ 0 | 1/ 0 | 0/ 0 | contCOC/ no |
| XXVII | 67.1 | Pain | 10/ n.a. | 10/ 0 | 10/ n.a. | 0/ 0 | 10/ n.a. | 6/ 0 | OP/ no |
| XXVIII | 72.2 | Pain | 8/ n.a. | 0/ 0 | 10/ n.a. | 3/ 0 | 8/ n.a. | 3/ 3 | contCOC/ contCOC |
| XXIX | 73.2 | Pain + Infertility | 6/ n.a. | 0/ 0 | 0/ n.a. | 7/ 0 | 8/ n.a. | 0/ 0 | no/ LRID |
| XXX | 76.3 | Umbilical bleeding | 0/ n.a. | 0/ 0 | 7/ n.a. | 0/ 0 | 7/ n.a. | 0/ 0 | cycCOC/ OP |
| XXXI | 79.1 | Pain + Infertility | 8/ 6 | 6/ 0 | 10/ 8 | 0/ 8 | 9/ 9 | 9/ 6 | no/ ESI |
| XXXII | 82.7 | Pain + Infertility | 9/ n.a. | 0/ 0 | 10/ n.a. | 0/ 0 | 8/ n.a. | 0/ 0 | no/ no |
| XXXIII | 88.6 | Pain | 0/ n.a. | 0/ 0 | 0/ n.a. | 0/ 4 | 0/ n.a. | 0/ 3 | contCOC/ contCOC |
| XXXIV | 94.1 | Pain + Infertility | 7/ 0 | 4/ 0 | 9/ 8 | 7/ 0 | 9/ 6 | 7/ 5 | no/ no |
| XXXV | 96.1 | Pain | 10/ n.a. | 0/ 0 | 10/ n.a. | 8/ 0 | 10/ n.a. | 6/ 0 | no/ no |
| XXXVI | 100.4 | Pain + Infertility | 0/ 0 | 0/ 0 | 8/ 0 | 0/ 0 | 8/ 0 | 5/ 0 | cycCOCP/ COCP |
| XXXVII | 104.9 | Pain | 10/ 6 | 7/ 6 | 10/ 8 | 10/ 6 | 10/ 7 | 10/ 7 | contCOC/ no |
| XXXVIII | 111.9 | Pain | 10/ n.a. | 0/ 0 | 10/ n.a. | 0/ 0 | 0/ n.a. | 5/ 0 | no/ estradiol |
| XXXIX | 113.8 | Imag | 2/ n.a. | 0/ 0 | 6/ n.a. | 0/ 2 | 4/ n.a. | 0/ 0 | contCOC/ OP |
| XL | 115.2 | Pain | 4/ n.a. | 0/ 0 | 8/ n.a. | 8/ 5 | 0/ n.a. | 0/ 0 | contCOC/ OP |
| XLI | 116.3 | Pain | 0/ 0 | 0/ 0 | 10/ 4 | 0/ 3 | 9/ 4 | 8/ 1 | contCOC/ no |
| XLII | 123.6 | Pain + Infertility | 6/ n.a. | 0/ 0 | 9/ n.a. | 5/ 0 | 9/ n.a. | 0/ 0 | cycCOC/ OP |
| XLIII | 123.8 | Pain | 8/ n.a. | 0/ 0 | 8/ n.a. | 6/ 2 | 0/ n.a. | 7/ 0 | contCOC/ OP |

F-Up: time of follow-up (months). Main complaints prior to surgery – Infertility: failure to achieve a clinical pregnancy after 12 months or more of regular unprotected sexual intercourse; Imaging: one or more important deep endometriosis lesions diagnosed by magnetic resonance imaging. M-Dysch: menstrual dyschezia. NM-Dysch: nonmenstrual dyschezia. Dysm: dysmenorrhea. DDysp: deep dyspareunia (deep pain during vaginal penetration). n.p.: no

*(Continued)*

**Table 4.** (Continued)

sexual intercourse. CLBPain: cyclic lower back pain. PPain: acyclic pelvic pain. The cyclical symptoms prior to surgery were assessed based on the periods when the patient was under no hormonal blockade. n.a.: not assessed (under hormonal blockade). LRID: levonorgestrel-releasing intrauterine device; cycCOC: cyclic combined oral contraceptive pill; contCOC: continuous combined oral contraceptive pill; OP: oral progesterone; ESI: etonogestrel subdermal implant. Case 3 presented umbilical hernia at follow-up assessment. Case 4 presented vaginal vault granulation at follow-up assessment.

major complication: a rectovaginal fistula (case XXXVII). In fact, after carefully reviewing the medical records, we strongly hypothesized that because the DCST was probably performed on a very low lesion (near the anal verge), the posterior vaginal wall was presumably included in the stapling, which would have "created " the fistula. Our results support the hypothesis that low intra- and postoperative severe complication rates are observed when the procedures are performed by experienced surgeons in an expert center [66].

In a recent systematic review with meta-analysis of observational studies of intestinal surgeries for the treatment of DE, the overall postoperative complications rate was 18.5%, and the most frequent complications considering all techniques was rectovaginal fistula (2.4%), followed closely by anastomotic leakage, a condition with high morbidity and mortality, which occurred in 2.1% of intestinal resections [67]. This meta-analysis's low reported complication rate is welcome, but as in any meta-analysis, the rate of complications and poor outcomes can be underestimated due to publication bias [68]. Our low rate of rectovaginal fistula at 2.3% (one case) was comparable to the rate reported in the meta-analysis, but should be considered outstanding as our series included mainly cases with very low lesions, which are notoriously harder to perform and entail a higher probability of complications. It is also important to note that in studies that demonstrated complications related to discoid bowel resection, the fistula rates were higher than in our cohort: 3.7-4.0%. Likewise, even the less invasive shaving technique presented a 2.1% risk of fistula.

Ileostomies and colostomies have been used widely, both in the treatment of bowel complications and even prophylactically in high-risk anastomosis. We have elected to not perform ostomies concomitantly with the DCST, a surgical strategy that appears to have been beneficial for our patients.

Although, in theory, the air insufflation test (used in this series) can detect bowel leakage during surgery, they do not provide direct observation of anastomosis quality from within the luminal cavity. Therefore, performing an intraoperative proctosigmoidoscopy (a feasible and non–time-consuming intraoperative procedure) could be considered to detect bowel leakage after discoid resection for rectosigmoid endometriosis [69].

### Pelvic organ dysfunction after surgery (a well-known risk)

Surgeries seeking complete resection of DE are among the most challenging for pelvic surgeons, as they carry multiple risks. Pelvic organ dysfunctions (urinary, defecatory and sexual) remain a point of concern even among the most experienced surgeons. Most endometriosis patients are young women, who are still aspire to become pregnant. Even though being treated for what is often considered a "benign disease," many women suffer greater personal, family and social impacts with these types of functional complications when compared, for example, to older patients undergoing cancer surgery in whom such complications are expected and thus more acceptable.

When bowel involvement is present, the segmental technique, which requires greater dissection and manipulation of the pelvis, presents the greatest risk of nerve damage with transitory or permanent consequences. As surgeons increasingly pursue complex deep endometriotic lesions infiltrating not only the lateral compartments [7] but also the pelvic floor musculature [8], nerve damage may occur regardless of the technique, even when no intestinal resection is performed [16].

Urinary dysfunction after surgery for the treatment of DE is a topic that always concerns medical teams and patients, particularly the prospect of definitive loss of bladder sensation or the need for long-term intermittent self-catheterization [16]. In general, the risk of urinary retention in intestinal resections is about 2% [67], but the greater the surgical radicality

**Table 5. Bowel habits prior to double circular stapler bowel surgery and at follow-up (prior to/ at follow-up).**

| Case | F-Up | QoLife | Cyclical changes | Cyclic bloating | Cyclic hematochezia | Laxatives | Duration | Incomplete | Interval | Anal disorders |
|---|---|---|---|---|---|---|---|---|---|---|
| I | 1.4 | poor/ poor | no/ n.a. | no/ n.a. | no/ n.a. | yes/ yes | NR/ NR | yes/ yes | NR/ NR | no/ no |
| II | 1.5 | poor/ good | diarrhea/ n.a. | yes/ n.a. | no/ n.a. | no/ no | 5/ 5 | no/ no | 72/ 24 | no/ no |
| III | 1.6 | good/ good | diarrhea/ n.a. | yes/ n.a. | no/ n.a. | no/ no | 10/ 5 | no/ no | 24/ 24 | no/ no |
| IV | 2.5 | excellent/ medium | no/ no | yes/ yes | no/ no | yes/ no | 4/ 40 | no/ yes | 24/ 24 | yes/ yes |
| V | 2.6 | terrible/ good | constipation/ diarrhea | yes/ yes | no/ no | no/ no | 30/ 10 | yes/ yes | 120/ 24 | no/ no |
| VI | 4.0 | medium/ good | diarrhea/ n.a. | no/ n.a. | no/ n.a. | no/ no | 5/ 5 | yes/ no | 24/ 48 | yes/ yes |
| VII | 7.4 | poor/ poor | constipation/ n.a. | yes/ n.a. | no/ no | no/ no | 5/ 30 | no/ yes | 24/ 72 | no/ no |
| VIII | 13.7 | good/ good | no/ no | yes/ yes | yes/ no | no/ no | 5/ 10 | no/ no | 24/ 24 | no/ no |
| IX | 15.1 | medium/ excellent | diarrhea/ no | yes/ yes | yes/ no | no/ no | 3/ 3 | no/ no | 12/ 12 | yes/ yes |
| X | 15.4 | medium/ medium | constipation/ no | yes/ no | no/ no | no/ no | 5/ 15 | no/ no | 24/ 24 | no/ no |
| XI | 16.1 | terrible/ good | diarrhea/ n.a. | yes/ n.a. | yes/ no | no/ no | 2/ 10 | no/ yes | 48/ 48 | no/ no |
| XII | 16.1 | medium/ excellent | diarrhea/ n.a. | yes/ n.a. | no/ no | no/ no | 5/ 2 | no/ no | 24/ 24 | yes/ no |
| XIII | 16.6 | medium/ good | constipation/ n.a. | yes/ n.a. | no/ no | no/ no | 15/ 10 | yes/ no | 12/ 12 | no/ no |
| XIV | 18.7 | poor/ good | diarrhea/ diarrhea | yes/ yes | no/ no | no/ no | 15/ 10 | yes/ no | 12/ 12 | no/ no |
| XV | 19.7 | terrible/ medium | diarrhea/ diarrhea | yes/ yes | no/ no | no/ no | 6/ 3 | yes/ yes | 96/ 24 | no/ no |
| XVI | 20.4 | good/ good | constipation/ n.a. | yes/ no | no/ no | no/ no | 2/ 2 | no/ no | 24/ 24 | no/ no |
| XVII | 20.9 | poor/ excellent | diarrhea/ n.a. | yes/ no | no/ no | no/ no | 10/ 5 | no/ no | 8/ 24 | no/ no |
| XVIII | 21.0 | poor/ good | diarrhea/ no | yes/ no | no/ no | no/ no | 15/ 2 | no/ no | 96/ 24 | yes/ no |
| XIX | 21.4 | terrible/ good | diarrhea/ no | yes/ yes | no/ no | no/ no | 10/ 5 | no/ no | 24/ 24 | yes/ yes |
| XX | 21.8 | good/ medium | no/ constipation | yes/ yes | no/ no | no/ no | 5/ 10 | no/ yes | 48/ 72 | yes/ yes |
| XXI | 37.7 | medium/ terrible | no/ constipation | yes/ yes | yes/ no | no/ yes | 5/ NR | no/ yes | 48/ > 168 | yes/ yes |
| XXII | 38.2 | terrible/ terrible | constipation/ constipation | no/ no | yes/ yes | no/ no | 20/ 20 | no/ no | 96/ 96 | no/ no |
| XXIII | 43.8 | terrible/ good | diarrhea/ diarrhea | yes/ yes | no/ no | no/ no | 10/ 2 | no/ no | 96/ 48 | no/ no |
| XXIV | 45.6 | poor/ poor | diarrhea/ diarrhea | yes/ yes | no/ no | yes/ yes | 30/ 30 | yes/ yes | 168/ 168 | no/ no |
| XXV | 48.2 | terrible/ good | diarrhea/ no | yes/ no | no/ no | no/ no | 30/ 5 | yes/ no | 72/ 24 | no/ no |
| XXVI | 66.1 | poor/ poor | diarrhea/ constipation | yes/ yes | no/ no | no/ no | 10/ 4 | no/ yes | 12/ 72 | no/ no |
| XXVII | 67.1 | terrible/ good | constipation/ n.a. | yes/ n.a. | no/ no | yes/ no | 20/ 10 | no/ yes | 12/ 8 | no/ no |
| XXVIII | 72.2 | medium/ medium | diarrhea/ n.a. | no/ n.a. | no/ n.a. | no/ no | 20/ 10 | no/ yes | 48/ 12 | yes/ yes |
| XXIX | 73.2 | good/ good | constipation/ n.a. | yes/ n.a. | no/ n.a. | no/ no | 10/ 10 | no/ no | 8/ 24 | no/ no |
| XXX | 76.3 | good/ medium | diarrhea/ n.a. | yes/ n.a. | no/ no | no/ no | 2/ 3 | no/ no | 24/ 12 | yes/ no |
| XXXI | 79.1 | medium/ good | diarrhea/ diarrhea | yes/ yes | yes/ no | no/ no | 2/ 2 | yes/ yes | 96/ 48 | no/ no |
| XXXII | 82.7 | medium/ medium | diarrhea/ n.a. | yes/ n.a. | no/ n.a. | no/ no | 15/ 10 | no/ yes | 144/ 96 | no/ no |
| XXXIII | 88.6 | poor/ medium | constipation/ n.a. | yes/ n.a. | no/ no | yes/ no | 3/ 6 | no/ no | 72/ 12 | no/ no |
| XXXIV | 94.1 | terrible/ medium | constipation/ constipation | yes/ yes | no/ no | no/ yes | 20/ 10 | yes/ yes | 120/ 72 | no/ no |
| XXXV | 96.1 | terrible/ excellent | diarrhea/ n.a. | yes/ n.a. | yes/ no | no/ no | 1/ 3 | yes/ no | 6/ 24 | no/ no |
| XXXVI | 100.4 | excellent/ excellent | no/ no | yes/ no | no/ no | no/ no | 5/ 5 | no/ no | 24/ 24 | no/ no |
| XXXVII | 104.9 | terrible/ poor | constipation/ constipation | yes/ yes | yes/ no | yes/ no | 15/ 10 | yes/ yes | 144/ 24 | yes/ yes |
| XXXVIII | 111.9 | poor/ medium | constipation/ n.a. | yes/ n.a. | no/ no | no/ no | 10/ 5 | no/ no | 48/ 48 | no/ no |
| XXXIX | 113.8 | poor/ good | diarrhea/ n.a. | yes/ n.a. | no/ n.a. | no/ no | 10/ 10 | no/ no | 48/ 24 | yes/ yes |
| XL | 115.2 | excellent/ excellent | diarrhea/ n.a. | yes/ n.a. | no/ n.a. | no/ no | 2/ 2 | no/ no | 8/ 12 | no/ yes |
| XLI | 116.3 | poor/ good | diarrhea/ diarrhea | yes/ yes | no/ no | no/ no | 10/ 3 | yes/ no | 48/ 24 | yes/ no |
| XLII | 123.6 | poor/ poor | diarrhea/ n.a. | yes/ n.a. | no/ no | no/ no | 5/ 15 | no/ yes | 24/ 48 | no/ yes |
| XLIII | 123.8 | good/ good | constipation/ n.a. | yes/ n.a. | yes/ no | no/ no | 3/ 5 | no/ no | 24/ 24 | no/ no |

F-Up: time of follow-up (months). QoLife: self-reported quality of life related to bowel function (terrible, poor, medium, good or excellent). n.a.: not assessed (under hormonal blockade). Cyclical changes: tendency to diarrhea or constipation that occurs cyclically. Bloating: reported as feeling full, tight,

*(Continued)*

**Table 5.** (Continued)

or swollen in the abdomen. Laxatives: regular use of any laxative. NR: not reported. Duration: estimated duration of evacuation (min). Incomplete: feeling of incomplete rectal evacuation (feeling that after finishing a defecation, still there are more stool that need to be eliminated). Interval: estimated interval without evacuation (hours). One woman (case XXI) became very constipated after the surgery, only being able to evacuate with the use of laxative" (interval between evacuations > 1 week). No patient had inflammatory bowel disease. Anal disorders (hemorrhoids, abscesses, fissures, cracks, fistulas and cancer) were also included because may affect the assessment of the bowel symptoms. Menstrual and nonmenstrual dyschezia assessments were presented in **Table 4**.

the greater the chance of some urinary dysfunction – especially when intestinal resection or bilateral manipulation of the parametria (lateral compartments) is required.

Happily, in our series no patient needed a Foley catheter for more than 90 days, but many developed transient unpleasant urinary symptoms, such as urgency, intermittency and straining to void (**Table 3**). Even though severe urinary disorders have been increasingly rare when nerve-sparing surgery is performed, every patient should be advised about the possibility of urinary changes and the potential need for long-term physical therapy after surgery.

In this cohort, the occurrence of a worsening of constipation in several cases can not be overlooked. Patients with severe preoperative constipation are less likely to achieve normal bowel movements after surgery for rectal DE, using either radical or conservative rectal procedures [70,71]. Some patients in our series may have reported an improvement in their quality-of-life related to bowel function despite the persistence of symptoms of mild impaired defecation, perhaps confounded with an overall improvement in pelvic pain. Indeed, the majority of the patients reported an important improvement in pelvic pain scores regardless of the duration of follow-up (**Table 4**).

## Size of the bowel endometriotic lesion and its distance from the anal verge

In the present study DCST was a safe and effective alternative to a conventional low segmental colorectal resection, with its associated higher risk of leakage due to the lack of a serosal layer [72]. Whenever possible a less extensive procedure should be chosen to avoid complications associated with more radical surgery. Indeed, the main advantage of the DCST is that it has replaced the most hazardous segmental colorectal resections, particularly very low bowel lesions less than 10 cm from the anal verge.

There is a growing interest in reporting the dimensions of endometriotic lesions in the rectosigmoid and their distance from the anal verge in studies of bowel DE resection [45]. However, although some authors have even used millimeter precision to assess this distance [73], the accuracy of these measurements are dubious due to both the limitations of imaging techniques and the elastic properties of the intestine. Similar concerns also apply to the surgical specimens sent for anatomopathological examination. Indeed, there are similar challenges in measuring the dimensions of the endometriotic lesion during laparoscopic surgery as well as estimating its distance from the anal verge at the beginning of the surgery (before rectal dissection) or immediately after the second stapling (when the rectosigmoid has been surgically released). As it pertains to the surgical planning, we believe that the size of the endometriotic lesion infiltrating the bowel measured by any imaging technique may not be a reliable predictor of the actual size of the intestinal lesion to be resected because part of the lesion often remains attached to other structures (such as the retrocervical area) after a complete laparoscopic dissection of the affected bowel segment and must be resected separately.

Finally, the DCST is not a good option for endometriotic lesions that are more than 18 cm above the anal verge as this is the furthest that the stapler can reach. Nor is it a good option for endometriotic lesions less than 7 cm from the anal verge because of the difficulty of averting inclusion of the vaginal mucosa in the stapling [43,45]. Actually, in this series, the option of performing DCST was defined during surgery, that is, observing the lesion after surgical dissection and taking into account the size of the stapler itself. The stapler device is introduced the necessary distance in accordance with the location of the endometriotic lesion so that the maneuvers necessary for stapling just outside the margins of the lesion are possible. After

**Table 6. Previous pelvic surgeries (medical history) and reproductive outcomes prior to and after double circular stapler bowel surgery (prior to/ at follow-up).**

| Case | F-Up | Previous surgery | Infert | Partner | Prior to surgery | | | At follow-up | | | |
|------|------|------|--------|---------|---------|-------|-----|---------|-------|-----|----------|
| | | | | | Pregnancy | G-P-A | V-C | Pregnancy | G-P-A | V-C | Surgeries |
| I | 1.4 | no | | yes/ yes | naturally | 3-2-1 | 0-2 | n.a. (hysterec) | n.a. | n.a. | no |
| II | 1.5 | endo | yes | no/ no | ART | 1-1-0 | 0-1 | did not try | n.a. | n.a. | no |
| III | 1.6 | no | | no/ no | naturally | 1-0-1 | 0-0 | n.a. (hysterec) | n.a. | n.a. | no |
| IV | 2.5 | no | | never/ never | did not try | 0-0-0 | 0-0 | did not try | 0-0-0 | 0-0 | no |
| V | 2.6 | no | | yes/ yes | did not try | 0-0-0 | 0-0 | did not try | 0-0-0 | 0-0 | no |
| VI | 4.0 | no | | no/ no | naturally | 2-1-1 | 0-1 | n.a. (hysterec) | 0-0-0 | 0-0 | no |
| VII | 7.4 | subtotal hysterec | | yes/ yes | naturally | 3-1-2 | 1-0 | n.a. (hysterec) | 0-0-0 | 0-0 | no |
| VIII | 13.7 | no | yes | yes/ yes | failed | 0-0-0 | 0-0 | failed | 0-0-0 | 0-0 | no |
| IX | 15.1 | no | | yes/ yes | did not try | 0-0-0 | 0-0 | naturally (pregnant) | 1-0-0 | 0-0 | no |
| X | 15.4 | no | yes | yes/ yes | failed | 0-0-0 | 0-0 | failed | 0-0-0 | 0-0 | no |
| XI | 16.1 | endo | | never/ never | did not try | 0-0-0 | 0-0 | did not try | 0-0-0 | 0-0 | no |
| XII | 16.1 | no | | never/ never | did not try | 0-0-0 | 0-0 | did not try | 0-0-0 | 0-0 | no |
| XIII | 16.6 | unioophorec | | yes/ yes | ectopic | 1-0-1 | 0-0 | did not try | 0-0-0 | 0-0 | no |
| XIV | 18.7 | no | | yes/ yes | did not try | 0-0-0 | 0-0 | naturally | 1-1-0 | 0-1 | no |
| XV | 19.7 | endo (twice) | yes | yes/ yes | naturally | 3-1-2 | 1-0 | didnottry | 0-0-0 | 0-0 | no |
| XVI | 20.4 | no | yes | yes/ yes | naturally | 2-2-0 | 0-2 | didnottry | 0-0-0 | 0-0 | no |
| XVII | 20.9 | no | | no/ yes | didnottry | 0-0-0 | 0-0 | didnottry | 0-0-0 | 0-0 | no |
| XVIII | 21.0 | no | | yes/ yes | naturally | 1-0-1 | 0-0 | naturally (pregnant) | 1-0-0 | 0-0 | no |
| XIX | 21.4 | no | | never/ never | didnottry | 0-0-0 | 0-0 | didnottry | 0-0-0 | 0-0 | no |
| XX | 21.8 | no | yes | yes/ yes | failed | 0-0-0 | 0-0 | failed | 0-0-0 | 0-0 | unisalpingec |
| XXI | 37.7 | hysterec | | yes/ yes | naturally | 2-2-0 | 0-2 | n.a. (hysterec) | 0-0-0 | 0-0 | no |
| XXII | 38.2 | no | | yes/ yes | naturally | 1-1-1 | 0-1 | Naturally | 1-1-0 | 0-1 | tuballigation |
| XXIII | 43.8 | appendec | yes | yes/ yes | naturally | 1-0-1 | 0-0 | failed | 0-0-0 | 0-0 | no |
| XXIV | 45.6 | unioophorec | | no/ yes | failed | 0-0-0 | 0-0 | didnottry | 0-0-0 | 0-0 | no |
| XXV | 48.2 | no | yes | yes/ yes | ART | 1-0-1 | 0-0 | naturally | 1-0-1 | 0-0 | no |
| XXVI | 66.1 | no | | never/ never | didnottry | 0-0-0 | 0-0 | didnottry | 0-0-0 | 0-0 | no |
| XXVII | 67.1 | no | | yes/ yes | didnottry | 0-0-0 | 0-0 | n.a. (hysterec) | 0-0-0 | 0-0 | # |
| XXVIII | 72.2 | no | | never/ yes | didnottry | 0-0-0 | 0-0 | naturally | 1-1-0 | 1-0 | no |
| XXIX | 73.2 | no | yes | yes/ yes | ART (4) | 0-0-0 | 0-0 | failed | 0-0-0 | 0-0 | no |
| XXX | 76.3 | no | | yes/ yes | didnottry | 0-0-0 | 0-0 | naturally | 1-1-0 | 0-1 | no |
| XXXI | 79.1 | no | yes | yes/ yes | failed | 0-0-0 | 0-0 | naturally | 2-2-0 | 2-0 | no |
| XXXII | 82.7 | endo | yes | yes/ yes | ART (2) | 0-0-0 | 0-0 | didnottry | 0-0-0 | 0-0 | no |
| XXXIII | 88.6 | no | | yes/ yes | naturally | 1-1-0 | 0-1 | didnottry | 0-0-0 | 0-0 | no |
| XXXIV | 94.1 | endo | yes | yes/ yes | failed | 0-0-0 | 0-0 | ART | 2-0-2 | 0-0 | myomec + endo |
| XXXV | 96.1 | hysterec + unioophorec | | yes/ yes | naturally | 1-1-0 | 0-1 | n.a. (hysterec) | 0-0-0 | 0-0 | no |
| XXXVI | 100.4 | no | yes | yes/ yes | failed | 0-0-0 | 0-0 | failed (ART) | 0-0-0 | 0-0 | no |
| XXXVII | 104.9 | no | | never/ yes | didnottry | 0-0-0 | 0-0 | ectopic (1) + failed (ART) | 1-0-1 | 0-0 | myomec + ecto |
| XXXVIII | 111.9 | endo | | yes/ yes | ART (4); naturally (1) | 2-2-0 | 0-2 | n.a. (hysterec) | 0-0-0 | 0-0 | hysterec + ooforec |
| XXXIX | 113.8 | no | | yes/ yes | didnottry | 0-0-0 | 0-0 | didnottry | 0-0-0 | 0-0 | no |

*(Continued)*

**Table 6.** (Continued)

| Case | F-Up | Previous surgery | Infert | Partner | Prior to surgery | | | At follow-up | | | Surgeries |
| | | | | | Pregnancy | G-P-A | V-C | Pregnancy | G-P-A | V-C | |
|---|---|---|---|---|---|---|---|---|---|---|---|
| XL | 115.2 | no | | yes/ yes | didnottry | 0-0-0 | 0-0 | naturally | 2-1-1 | 0-1 | no |
| XLI | 116.3 | no | | yes/ yes | didnottry | 0-0-0 | 0-0 | naturally | 4-2-2 | 0-2 | no |
| XLII | 123.6 | no | yes | yes/ yes | failed | 0-0-0 | 0-0 | naturally | 1-1-0 | 0-1 | no |
| XLIII | 123.8 | endo | yes | no/ yes | failed | 0-0-0 | 0-0 | ART | 1-1-0 | 0-1 | no |

F-Up: time in months since surgery at last follow-up. Hysterec: hysterectomy; Uni: unilateral; Oophorec: oophorectomy; Endo: endometriosis; Appendec: appendectomy; Salpingec: salpingectomy; Myomec: myomectomy; Ecto: ectopic pregnancy. Infert: infertility (failure to achieve a clinical pregnancy after 12 months or more of regular unprotected sexual intercourse). Partner: prior to surgery/ at follow-up. G: number of gestations. P: number of births. A: number of abortions. V-C: vaginal delivery (n)/ Caesarean-section (n). ART: assisted reproductive technology. Cases IX and XVIII were pregnant at the follow-up assessment. (#) Case XXVII: Two years after underwent double circular stapler technique (DCST) for colorectal deep endometriosis, she underwent another laparoscopic endometriosis surgery, which included segmental colorectal resection, hysterectomy, unilateral oophorectomy and pelvic lymphadenectomy; the length of permanence of the Foley catheter after surgery was 30 days.

reviewing most surgeries and discussing their estimated distances, we conservatively concluded that the lesions resected in this series, with the exception of the case V, were located between 7 and 18 cm from the anal verge (**Table 2**).

The double circular stapler technique allows excision of macroscopically visible lesions up to 5 cm in extension. There is always the possibility that "positive margins" – endometriotic lesions that were not visually perceived during the minimally invasive procedure – will be identified during microscopic analysis of the removed specimens [45]. This issue was not addressed in this study, since it is not known what would be the clinical impact of permanence microscopic endometriotic foci in cytoreductive surgeries for the treatment of deep endometriosis.

## Choosing the bowel resection technique

Because it is not an oncological disease, endometriosis requires – or would could say "allows" – a less "aggressive" bowel resection. While there is a natural desire to resect all the identified foci of deep infiltrating endometriosis both to eliminate the cause of symptoms and to reduce the chance of recurrence, the surgery should be performed in the least radical way possible to minimize the complications and sequelae of the surgery itself. In addition to highlighting the need for a nerve preserving surgical act to prevent complications during colorectal endometriosis [18,74], it is important to consider potential helpful procedures that can be adopted, such as the use of endovenous indocyanine green [75] - strategy not used in this case series.

Although the definitive diagnosis of deep endometriosis still requires histological confirmation, diagnostic imaging methods permit quite precise mapping of likely endometriotic lesions enabling good surgical planning. However, two statements about intestinal DE should always be remembered: 1) it is practically impossible to define preoperatively the exact bowel resection technique that will be performed; and 2) the actual extent of the intestinal lesions can only be assessed after surgical systematization and thorough pelvic adhesiolysis, which "restore" the pelvic anatomy. Therefore, multidisciplinary surgical teams must be trained and prepared to perform any of the available techniques according to the characteristics (extension, depth, circumferential extension, etc.) of the lesions.

As previously mentioned, three main techniques for treating intestinal endometriotic lesions are used: shaving, discoid resection and segmental resection. The DCST is about avoiding a segmental resection in order to reduce the risk of functional impairments potentially caused by surgery [17]. As much as conservative techniques have already been described and published internationally, DCST seems to be still underused, most likely due to the lack of studies that include long-term follow-up. Finally, there is another procedure, the linear stapler nodulectomy, which has promising preliminary results, and may be able to obviate the need for segmental resection of some lesions, especially nodules smaller than 3 cm [63].

## Limitations and strengths

### Design limitations

Perhaps, the main limitation in this study is inherent to its nature, that is, its design. As an uncontrolled observational study, it allows the presence of covariates that make interpretation of the findings and objective conclusions difficult. Nevertheless, to our knowledge, this is the largest published study of the long-term follow-up of DCST for bowel endometriosis resection.

While most follow-up assessments were prospective, this study is essentially a retrospective analysis of an intervention based on chart reviews and included data abstracted from medical records of surgical patients. Still, the present study may be considered stronger than most retrospective studies because it relied on systematic data-gathering processes instituted over 10 years ago and expanded and refined since then, carried out consistently by clinician-researchers and institute leadership who were committed to outcomes research from the outset.

Although the number of DE cases does not lend itself to estimate of the risk of surgical complications with a very precise confidence interval, the results reported in this study represent an institutional experience over an extended period that support the DCST as a safe alternative to segmental colorectal resection in several specific clinical situations.

### What we were not able to assess

In addition to individual pelvic pain and infertility responses – the major endometriosis-related complaints, this study included bowel and bladder dysfunctions as functional outcomes that are relevant in bowel surgery for DE [16,76]. Although the long-term functional outcomes relevant to bowel surgery for DE ideally should also include a complete sexual function assessment encompassing desire, arousal, lubrication, stimulation, satisfaction [77] - as this also impacts overall quality-of-life [76,78,79], we did not assess sexual function in a systematic manner.

In the present study, we did not consider the margins because the specimens were not microscopically assessed in all patients in a systematic and reliable manner for research purposes. In fact, it has already been shown that even in the most extensive segmental colorectal resections, margins can be compromised [80], and there is always the possibility that positive margins will be seen at microscopic analysis of the removed specimens [45].

We considered the common use of measures of central tendency (e.g., mean or median) to assess self-rated scores (e.g., pain quantification) or calculated indices (e.g., scored questionnaires) as a point of concern in some endometriosis studies assessing surgical outcomes because they usually do not discuss the nature of the variables [81], that is, whether a variable is a ratio or ordinal and is linear or nonlinear [2]. Thus, instead of presenting and discussing measures of central tendency, we elected to present the characteristics of each patient and multiple functional outcomes for each of the 43 cases. This enables the reader to explore the particularities of each case-"real life as it is". Moreover, individual raw data offers readers unfamiliar with such instruments a better understanding of the complexity of deep infiltrating endometriosis from a more qualitative perspective.

### Learning curve

The fundamental goal of nerve-sparing pelvic surgery is to minimize damage to the pelvic innervation in order to minimize postoperative vaginal/sexual, bladder, rectal dysfunction [36,82]. Nerve-sparing surgical techniques naturally have evolved and been refined since 2010, and indeed our team's minimally invasive approach via avascular spaces minimizes both bleeding and injury to pelvic splanchnic nerves. In this vein, we recognize experience and refinement of surgical technique constitute "a learning curve" and there should be a tendency towards fewer complications and better outcomes over time in the present study.

### Considerations about bias and external validity

Because some preoperative data needed to be obtained retrospectively, this study was prone to recall bias, or more precisely a "retrieval bias," particularly for cases performed before January 2018. This potential measurement bias, also

called detection bias, refers to any systematic or non-random error that may have occurred in some cases during the chart abstraction process.

The possibility of selection bias associated with access to care should be considered in this study. Thus, detailed characteristics of each member of the cohort were presented so that the interpretation of findings can be made in the context of these attributes of cohort, and inform the limits of its generalizability.

The possibility of factors not associated with the DCST itself influencing one or more functional outcomes also should be considered. Among these confounding factors, we highlight the resection of deep lesions infiltrating richly innervated structures, such as the lateral compartments of the pelvis, and the considerable heterogeneity across the 43 cases in terms of the constellation of lesions infiltrating multiple organs or different areas of the pelvic anatomy.

Unfortunately, minimal considerations concerning psychological aspects of women's health could not be included.

### Future studies

The findings of this study may serve as a starting point, helping to formulate hypotheses to be tested in future prospective studies.

"Bloating," a subjective discomfort characterized by the patient's sensation of intestinal gas, and cyclic abdominal distension with a measurable increase in abdominal girth [83] are two additional symptoms associated with bowel invasive endometriosis that were not explicitly addressed in this study. An assessment of changes in cyclical abdominal distention and subjective bloating after intestinal surgery for DE could be included in future prospective studies. All future studies should more systematically incorporate data and correlate symptoms about whether the subject received hormonal therapy both before and after surgery.

Although complete surgical microscopic removal may be a challenging goal [84] and the presence of positive resection margins does not seem to influence the clinical outcomes of segmental colorectal resections [76], we consider the main important conceptual concern, especially in younger patients, with reproductive aspirations who can potentially be referred for assisted reproduction techniques. This issue should be considered in future studies investigating recurrence of the disease and symptoms.

Regarding future before-and-after studies, our team recommends being very cautious when comparing median (or mean) scores because endometriosis is a highly-individualized condition, as are results and adverse consequences of its treatment [2,8,52].

### Conclusion

Preliminary results were encouraging in the past and the current assessment including long-term follow-up supports DCST for rectosigmoid deep endometriosis resection as a feasible, useful and safe strategy for avoiding segmental colorectal resection, particularly endometriotic nodules close to the anal verge.

### Supporting information

**S1 Table. Raw data.**
(XLSX)

### Acknowledgments

The authors thank Dr. Leigh J. Passman for reviewing the English manuscript.

### Author contributions

**Conceptualization:** Claudio Peixoto Crispi Jr, Bruna Rafaela Santos de Oliveira, Marlon de Freitas Fonseca.

**Data curation:** Claudia Maria Vale Joaquim, Nilton de Nadai Filho, Bruna Rafaela Santos de Oliveira, Camilla Gabriely Souza Guerra, Marlon de Freitas Fonseca.

**Formal analysis:** Claudio Peixoto Crispi Jr, Marlon de Freitas Fonseca.

**Investigation:** Claudio Peixoto Crispi Jr, Bruna Rafaela Santos de Oliveira, Camilla Gabriely Souza Guerra.

**Methodology:** Claudio Peixoto Crispi Jr, Claudio Peixoto Crispi, Claudia Maria Vale Joaquim, Paulo Sergio da Silva Reis Jr, Nilton de Nadai Filho, Marlon de Freitas Fonseca.

**Project administration:** Claudio Peixoto Crispi, Marlon de Freitas Fonseca.

**Software:** Marlon de Freitas Fonseca.

**Supervision:** Claudio Peixoto Crispi, Marlon de Freitas Fonseca.

**Validation:** Marlon de Freitas Fonseca.

**Visualization:** Claudia Maria Vale Joaquim, Paulo Sergio da Silva Reis Jr, Nilton de Nadai Filho, Bruna Rafaela Santos de Oliveira, Camilla Gabriely Souza Guerra, Marlon de Freitas Fonseca.

**Writing – original draft:** Claudio Peixoto Crispi Jr, Nilton de Nadai Filho, Marlon de Freitas Fonseca.

**Writing – review & editing:** Nilton de Nadai Filho, Marlon de Freitas Fonseca.

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
