## [Decision Letter · Decision Letter 0]

29 Jan 2025

PONE-D-24-56141Follow-up of bowel endometriosis resections performed using the Double Circular Stapler Technique: a decade’s experiencePLOS ONE

Dear Dr. Crispi Jr,

Thank you for submitting your manuscript to PLOS ONE. After careful consideration, we feel that it has merit but does not fully meet PLOS ONE’s publication criteria as it currently stands. Therefore, we invite you to submit a revised version of the manuscript that addresses the points raised during the review process.

We look forward to receiving your revised manuscript.

Kind regards,

Diego Raimondo

Academic Editor

PLOS ONE

Journal Requirements:

Reviewers' comments:

Reviewer's Responses to Questions

**Comments to the Author**

1. Is the manuscript technically sound, and do the data support the conclusions?

Reviewer #1: Yes

2. Has the statistical analysis been performed appropriately and rigorously? 

Reviewer #1: I Don't Know

3. Have the authors made all data underlying the findings in their manuscript fully available?

Reviewer #1: Yes

4. Is the manuscript presented in an intelligible fashion and written in standard English?

Reviewer #1: Yes

5. Review Comments to the Author

Reviewer #1: This study on double discoid resection for intestinal endometriosis is interesting, especially given the results reported. In fact, post-operative complications such as post-operative strictures of the intestinal suture and immediate post-operative bleeding are usually reported with a higher incidence in the case series of various authors. However, there are some aspects to explore further. In particular, considering the time that has passed since the first revision of the article, there has been an important scientific production on the author's topic and therefore the manuscript should be implemented in the discussion.

-In the discussion the author reports that it is not possible to give an indication of the technique based only on diagnostic imaging, but then what is the dimensional criterion in his series for deciding on a double discoid resection? What are the medians for the size of the nodules in your series?

-line 544-45: I agree with your statement, but it must be discussed in light of the updated available literature in this regard. I recommend evaluating a recent work by the group of Ianieri et al. (Anatomical-based classification of dorsal parametrectomy for deep endometriosis. Correlation with surgical complications and functional outcomes: A single-center prospective study. doi: 10.1002/ijgo.15781).

-also it would be interesting to comment in your discussion on a recent work on discoid intestinal resections in which an intraoperative rectoscopy was systematically used to reduce the risk of subsequent complications (Feasibility of Intraoperative Proctosigmoidoscopy After Discoid Bowel Resection for Deep Infiltrating Endometriosis: A Pilot Multicenter Study.

doi: 10.1016/j.jmig.2024.05.004).Do you think it could be of help given the technique used in your case study?

6. PLOS authors have the option to publish the peer review history of their article (what does this mean? ). If published, this will include your full peer review and any attached files.

**Do you want your identity to be public for this peer review?** For information about this choice, including consent withdrawal, please see our Privacy Policy .

Reviewer #1: No

---

## [Author Response · Author response to Decision Letter 1]

12 Feb 2025

Reviewer #1: This study on double discoid resection for intestinal endometriosis is interesting, especially given the results reported. In fact, post-operative complications such as post-operative strictures of the intestinal suture and immediate post-operative bleeding are usually reported with a higher incidence in the case series of various authors. However, there are some aspects to explore further. In particular, considering the time that has passed since the first revision of the article, there has been an important scientific production on the author's topic and therefore the manuscript should be implemented in the discussion.

Authors reply:

First, we would like to thank the reviewer for this insightful comment. Indeed, the problems that occurred in the last few months (our problems) delayed the submission process... and science does not allow delays. When doing a new search (February/09, 2025) using the keywords "double discoid / double circular", we identified 3 (three) publications, about which we were not aware before. Although none of the content of these articles contradicts our findings and conclusions, we considered them (as exposed below).

[1] The article published by Dabi et al. (2024) aimed “to study the impact of discoid excision for deep endometriosis with colorectal involvement on fertility outcomes”. Their study was present as "the first study to focus on fertility outcomes after discoid excision in patients with deep endometriosis and colorectal endometriosis". The authors state: "...The low intra- and postoperative severe complication rates observed in our cohort are probably because the procedures were performed by experienced surgeons." Also, although Double Discoid excision was performed only in 3 (three) of the 49 cases (6.1 % of the sample), their results support that discoid excision is safe and associated with good fertility outcomes.

We added this sentence at the end of the topic Surgical complications (Discussion Section):

"Our results support the hypothesis that low intra- and postoperative severe complication rates are observed when the procedures are performed by experienced surgeons in an expert center (Dabi et al., 2024)".

Dabi Y, Ebanga L, Favier A, Kolanska K, Puchar A, Jayot A, Ferrier C, Touboul C, Bendifallah S, Darai E. Discoid excision for colorectal endometriosis associated infertility: A balance between fertility outcomes and complication rates. J Gynecol Obstet Hum Reprod. 2024 Feb;53(2):102723. doi: 10.1016/j.jogoh.2024.102723. Epub 2024 Jan 9. PMID: 38211693.

[2] The video article (a case report) published by Hardman et al. (2024) aimed “to present the use of robotic-integrated ultrasound for performing a double discoid excision of multifocal rectosigmoid endometriosis”. Objectively, the technique used by the authors is completely different from ours: they used two circular staplers to treat small, independent bowel endometriotic lesions with a certain distance between them, but our technique uses two consecutive circular staplers to treat a single lesion that is larger than a single circular stapler could do. Therefore, we consider the inclusion of this reference unnecessary.

Hardman D, Bennett RD, Mikhail E. Fertility sparing double discoid excision of rectosigmoid deep endometriosis under robotic-integrated ultrasound guidance. Fertil Steril. 2024 Jan;121(1):126-127. doi: 10.1016/j.fertnstert.2023.10.001. Epub 2023 Oct 7. PMID: 37813274.

[3] The article published by Malzoni et al. (2023) is a “stepwise demonstration of the technique with narrated video footage"; it aimed “to show the surgical steps used to perform a laparoscopic double discoid colorectal resection for the excision of 2 distinct deep endometriotic nodules”. In their publication, once again, the "double discoid" technique was used to treat small (<3 cm) independent intestinal endometriotic lesions. Therefore, we consider the inclusion of this reference unnecessary.

[Obs. In our opinion, the terminology used for the treatment of small independent lesions (distant from each other) and for the treatment of a single larger lesion (not completely removable with a single load of circular stapling) should not be the same in the literature.]

Malzoni M, Coppola M, Casarella L, Iuzzolino D, Rasile M, Di Giovanni A, Falcone F. Laparoscopic Double Discoid Colorectal Resection for the Treatment of Two Distinct Deep Endometriotic Nodules. J Minim Invasive Gynecol. 2023 Dec;30(12):946-947. doi: 10.1016/j.jmig.2023.09.010. Epub 2023 Sep 24. PMID: 37748750.

-In the discussion the author reports that it is not possible to give an indication of the technique based only on diagnostic imaging, but then what is the dimensional criterion in his series for deciding on a double discoid resection? What are the medians for the size of the nodules in your series?

Authors reply:

We do appreciate your inquiry. This point needs to become clear.

As we mentioned in our manuscript, we think that the real size of the isolated bowel lesions can often be underestimated (or overestimated) by the tests traditionally used for preoperative endometriosis mapping - as described in the following sentence "...As it pertains to the surgical planning, we believe that the size of the endometriotic lesion infiltrating the bowel measured by any imaging technique may not be a reliable predictor of the actual size of the intestinal lesion to be resected because part of the lesion often remains attached to other structures (such as the retrocervical area) after a complete laparoscopic dissection of the affected bowel segment and must be resected separately..." (lines: 710-714). Following the premise that significant figures are the number of digits in a value that contribute to its accuracy, we did not present size measurements with millimeter precision in our study because the great difficulty of such precise measurement during the intraoperative period - as described in the following sentence "...the accuracy of these measurements are dubious due to both the limitations of imaging techniques and the elastic properties of the intestine. Similar concerns also apply to the surgical specimens sent for anatomopathological examination..." (lines: 706-707).

We made changes to the manuscript with the aim of clarifying how the size of the intestinal lesions/nodules were considered in the decision-making process of whether or not to perform a second stapling procedure following the first.

The sentences below were included in the topic Surgery - Materials and Methods Section:

“Regarding the dimensional criteria for deciding on a double discoid resection, these should be defined by the surgeon, since the size estimates obtained in preoperative imaging exams do not offer millimetric accuracy. In fact, the intestinal nodule is not measured during surgery, but rather its size is compared to the dimensions of the stapler (objectively, the groove created between the anvil and the stapler, when it is open). With a second circular stapler readily available for use, the surgeon performs the first stapling always with the aim of removing the largest part of the lesion (if possible, the entire nodule). Then, a second stapling is performed, if necessary.”

-line 544-45: I agree with your statement, but it must be discussed in light of the updated available literature in this regard. I recommend evaluating a recent work by the group of Ianieri et al. (Anatomical-based classification of dorsal parametrectomy for deep endometriosis. Correlation with surgical complications and functional outcomes: A single-center prospective study. doi: 10.1002/ijgo.15781).

Authors reply:

The article published by Ianieri et al. (2024) aimed "to evaluate complication rate and functional outcomes of nerve-sparing parametrectomy for deep endometriosis in relation to the extent of the surgical procedure, based on recognizable anatomical landmarks." In their article, “bladder voiding deficit occurred in 9.7% of cases, with higher rates in the deeper parametrectomy groups”.

Our study emphasizes the risks of pelvic dysfunctions after nerve-spare surgery, which should not be neglected in shared decision-making [please see the topic Pelvic organ dysfunction after surgery (a well-known risk) - Discussion Section].

Being very careful not to lose the main focus (bowel endometriosis resections performed using the Double Circular Stapler Technique), two sentences were added to the topic The lateral compartment (parametrial region) - Materials and Methods Section:

"Large resections in the parametrium (deep parametrectomy), even if carried out by expert surgeons, demonstrate a non-negligible rate of bladder voiding deficit (Ianieri at al., 2024). Furthermore, unilateral nerve preservation during parametrectomy is not sufficient to prevent persistent urinary retention after cytoreductive endometriosis surgery (Gasparoni et al., 2024).”

Ianieri MM, Alesi MV, Querleu D, Ercoli A, Chiantera V, Carcagnì A, Campolo F, Greco P, Scambia G. Anatomical-based classification of dorsolateral parametrectomy for deep endometriosis. Correlation with surgical complications and functional outcomes: A single- center prospective study. Int J Gynaecol Obstet. 2024 Dec;167(3):1043-1054. doi: 10.1002/ijgo.15781. Epub 2024 Jul 19. PMID: 39031095.

Gasparoni MP Jr, de Freitas Fonseca M, Favorito LA, da Silva Filho FS, Diniz ALL, Schuh MF, Gomes FH, de Resende JAD Jr. Unilateral nerve preservation during parametrectomy is not sufficient to prevent persistent urinary retention after cytoreductive endometriosis surgery. Arch Gynecol Obstet. 2024 Dec;310(6):3267-3278. doi: 10.1007/s00404-024-07842-2. Epub 2024 Nov 28. PMID: 39609310.

-also it would be interesting to comment in your discussion on a recent work on discoid intestinal resections in which an intraoperative rectoscopy was systematically used to reduce the risk of subsequent complications (Feasibility of Intraoperative Proctosigmoidoscopy After Discoid Bowel Resection for Deep Infiltrating Endometriosis: A Pilot Multicenter Study. doi: 10.1016/j.jmig.2024.05.004). Do you think it could be of help given the technique used in your case study?

Authors reply:

This question will certainly be a topic of discussion during our next surgeries. Thank you very much! We added this small paragraph at the end of the topic Surgical complications (Discussion Section):

“Although, in theory, the air insufflation test (used in this series) can detect bowel leakage during surgery, they do not provide direct observation of anastomosis quality from within the luminal cavity. Therefore, performing an intraoperative proctosigmoidoscopy (a feasible and non–time-consuming intraoperative procedure) could be considered to detect bowel leakage after discoid resection for rectosigmoid endometriosis (Raimondo et al., 2024).”

Raimondo D, Ianieri MM, Raffone A, Ferla S, Raspollini A, Virgilio A, Govoni F, Pavone M, Neola D, Guida M, Del Governatore M, Scambia G, Seracchioli R. Feasibility of Intraoperative Proctosigmoidoscopy After Discoid Bowel Resection for Deep Infiltrating Endometriosis: A Pilot Multicenter Study. J Minim Invasive Gynecol. 2024 Aug;31(8):680-687. doi: 10.1016/j.jmig.2024.05.004. Epub 2024 May 16. PMID: 38761918.

---

## [Editor Report · Decision Letter 1]

14 Feb 2025

Follow-up of bowel endometriosis resections performed using the Double Circular Stapler Technique: a decade’s experience

PONE-D-24-56141R1

Dear Dr. Crispi Jr,

We’re pleased to inform you that your manuscript has been judged scientifically suitable for publication and will be formally accepted for publication once it meets all outstanding technical requirements.

Kind regards,

Diego Raimondo

Academic Editor

PLOS ONE
---

## [Editor Report · Acceptance letter]

PONE-D-24-56141R1

PLOS ONE

Dear Dr. Crispi Jr,

I'm pleased to inform you that your manuscript has been deemed suitable for publication in PLOS ONE. Congratulations! Your manuscript is now being handed over to our production team.

Kind regards,

on behalf of

Dr. Diego Raimondo

Academic Editor

PLOS ONE